



# Dealing with Spatial Heterogeneity in Pointwise to Gridded Data Comparisons

Amir H. Souri[1*], Kelly Chance[1], Kang Sun[2,3], Xiong Liu[1], and Matthew S. Johnson[4]

[1]Atomic and Molecular Physics (AMP) Division, Harvard–Smithsonian Center for Astrophysics, Cambridge, MA, USA
[2]Department of Civil, Structural and Environmental Engineering, University at Buffalo, Buffalo, NY, USA
[3]Research and Education in Energy, Environment and Water Institute, University at Buffalo, Buffalo, NY, USA
[4]Earth Science Division, NASA Ames Research Center, Moffett Field, CA, USA

*Corresponding author: ahsouri@cfa.harvard.edu

**Abstract**

Atmospheric modelers and the trace gas retrieval community typically presuppose that pointwise measurements, which roughly represent the element of space, should compare well with satellite (model) pixels (grids). This assumption implies that the field of interest must possess a high degree of spatial homogeneity within the pixels (grids), which may not hold true for species with short atmospheric lifetimes or in the proximity of plumes. Results of this assumption often lead to a perception of a nonphysical discrepancy between data, resulting from different spatial scales, potentially making the comparisons prone to overinterpretation. Semivariogram is a mathematical expression of spatial variability in discrete data. Modeling the semivariogram behavior permits carrying out spatial optimal linear prediction of a random process field using kriging. Kriging can extract the spatial information (variance) pertaining to a specific scale, which in turn translating pointwise data to a grid space with quantified uncertainty such that a grid-to-grid comparison can be made. Here, using both theoretical and real-world experiments, we demonstrate that this classical geostatistical approach can be well adapted to solving problems in evaluating model-predicted or satellite-derived atmospheric trace gases. This study demonstrates that satellite validation procedures must take kriging variance and satellite spatial response functions into account. We present the comparison of Ozone Monitoring Instrument (OMI) tropospheric $NO_2$ columns against 11 Pandora Spectrometer Instrument (PSI) systems during the DISCOVER-AQ campaign over Houston. The least-squares fit to the paired data shows a low slope (OMI=0.76×PSI+1.18×$10^{15}$ molecules cm$^{-2}$, r$^2$=0.67) which is indicative of varying biases in OMI. This perceived slope, induced by the problem of spatial scale, disappears in the comparison of the convolved kriged PSI and OMI (0.96×PSI+0.66×$10^{15}$ molecules cm$^{-2}$, r$^2$=0.72) illustrating that OMI possibly has a constant systematic bias over the area. To avoid gross errors in comparisons made between gridded data versus pointwise measurements, we argue that the concept of semivariogram (or spatial auto-correlation) should be taken into consideration, particularly if the field exhibits a strong degree of spatial heterogeneity at the scale of satellite and/or model footprints.



## 1. Introduction

Most of the literature on validation of satellite trace gas retrievals or atmospheric chemical transport models assume that geophysical quantities within a satellite pixel or a model grid are spatially homogeneous. Nevertheless, it has long been recognized that this assumption can often be violated; spatially coarse atmospheric models or satellites are often not able to represent features, nor physical processes, transpiring at fine spatial scales. Janjic et al. [2016] used the term of *representation error* to describe this complication. They posit that this problem is a result of two combined factors: unresolved spatiotemporal scales and physiochemical processes. To elaborate on this definition, let us assume that an atmospheric model can represent the exact physiochemical processes but is fed with a constant $CO_2$ emission rate. This model obviously cannot resolve the spatial distribution of $CO_2$ concentration because we use an unresolved emission input. As another example, if we know the exact rates of $CO_2$ emissions but use a model unable to resolve atmospheric dynamics, the spatial distribution of $CO_2$ concentrations will be unrealistic due to unresolved physical processes.

Numerous scientific studies have reported on this matter. The simulations of short lifetime atmospheric compounds such as nitrogen dioxide ($NO_2$), isoprene, formaldehyde (HCHO), and the hydroxyl radical (OH) have been found to be strongly sensitive to the model spatial resolution [Vinken et al., 2011; Valin et al., 2011; Yu et al., 2016; Pan et al., 2017]. Likewise, the performance of weather forecast models in resolving non-hydrostatic components heavily relies on both model resolution and parametrizations used. For example, when Kendon et al. [2014], Souri et al. [2020a], and Wang et al. [2017] defined a higher spatial resolution grid in conjunction with more elaborate model physics, they were able to more realistically simulate extreme or local weather phenomena such as convection and sea-land breeze circulation.

The spatial representation issue is not only limited to models. Satellite trace gas retrievals optimize the concentration of trace gases and/or atmospheric states to best match the observed radiance using an optimizer along with an atmospheric radiative transfer model. This procedure requires various inputs such as surface albedo, cloud and aerosol optical properties, and trace gas profiles, all of which come with different scales and representation errors. Moreover, the radiative transfer model by itself has different layers of complexity with regards to physics. A myriad of studies have reported that satellite-derived retrievals underrepresent spatial variability whenever the prognostic inputs used in the retrieval are spatially unresolved [e.g., Russell et al., 2011; Laughner et al., 2018; Souri et al., 2016; Goldberg et al., 2019; Zhao et al., 2020]. Additionally, the large footprint of some sensors relative to the scale of spatial variability of species inevitably leads to some degree of the representativity issues [e.g., Souri et al., 2020b, Tang et al., 2021; Judd et al., 2020].

The validation of satellites or atmospheric models is widely done against pointwise measurements. Mathematically, a point is an element of space. Hence, it is not meaningful to associate a point with a spatial scale. If one compares a grid to a point sample, they are assuming that the point is the representative of the grid. At this point, the fundamental question is: is such a comparison ever logical, in the sense that the average of the spatial distribution of the underlying compound is represented by a single value measured at a subgrid location? This question was answered in Matheron [1963]. He advocated the notion of the semivariogram, a mathematical description of the spatial variability, which finally led to the invention of kriging, the best unbiased linear estimator of a random field. A kriging model can estimate a geophysical quantity in a common grid. This is not exclusively special; a simple interpolation method such as the nearest neighbor has the same purpose. The power of kriging lies in the fact that it takes the data-driven

spatial variability information into account and informs an error associated with the interpolated
map. This strength not only makes kriging a relatively superior model over simplified interpolation
methods, but also reflects the level of confidence pertaining to spatial heterogeneity dictated by
both data and the semivariogram model used through its variance [Chilès and Delfiner, 2009].
Different studies leveraged this classical geostatistical method to map the concentrations
of different atmospheric compounds at very high spatial resolutions [Tadíc et al., 2017; Li et al.,
2019; Zhan et al., 2018]; To the best of our knowledge, Swall and Foley, [2009] is the only study
that used kriging for a chemical transport model validation with respect to surface ozone. They
suggested that kriging estimation should be executed in grids rather than discrete points. Kriging
uses a semivariogram model in a continuous form. Optimizing the kriging grid size (i.e., domain
discretization) at which the estimation is performed is an essence to fully obtaining the maximum
spatial information from data. Another important caveat with Swall and Foley [2009] is that
averaging discrete estimates (points) to build grids is not applicable for remote sensing data.
Depending on the optics and the geometry, the spatial response function can transform from an
ideal box (simple average) to a sophisticated shape such as a super Gaussian function (weighted
average) [Sun et al., 2018]. Moreover, the footprint of satellites is not spatially constant. We will
address these complications in this study using both theoretical and real-world experiments.
Our paper is organized with the following sections. Sections 2 is a thorough review of the
concept of the semivariogram and kriging. We then provide different theoretical cases, their
uncertainty, sensitivities with respect to difference tessellation, grid size, and the number of
samples. Section 3 proposes a framework for satellite (model) validation using sparse point
measurements and elaborates on the representation error using idealized experiments. Sections 4
introduces several real-world experiments.
**2.   Semivariogram and Ordinary Kriging Estimator**
*2.1. Definition*

The semivariogram is a mathematical representation of the degree of spatial variability (or
similarity) in a function describing a regionalized geophysical quantity (*f*), which is defined as
[Matheron, 1963]:

$$\gamma(h) = \frac{1}{2V} \iiint_V [f(x+\boldsymbol{h}) - f(x)]^2 \, dV \tag{1}$$

where $x$ is a location in the geometric fields of $V$, $f(x)$ is the value of a quantity at the location of $x$,
and $\boldsymbol{h}$ is the vector of distance. If discrete samples are available rather than the continuous field,
the general formula can be simplified to the experimental semivariogram defined as:

$$\gamma(h) = \frac{1}{2N(\boldsymbol{h})} \sum_{|x_i - x_j| - |\boldsymbol{h}| \le \varepsilon} [Z(x_i) - Z(x_j)]^2 \tag{2}$$

where $Z$ is discrete observations (or samples), $N(\boldsymbol{h})$ is the number of paired observations separated
by the vector of $\boldsymbol{h}$. |.| operator indicates the length of a vector. The condition of $|x_i - x_j| - |\boldsymbol{h}| \le$
$\varepsilon$ is to allow certain tolerance for differences in the length of the vector. For simplicity, we only
focus on an isotropic case meaning we rule out the directional (or angular) dependency in $\gamma(h)$.
If a reasonable number of samples is present, one can describe $\gamma(h)$ through a regression
model (e.g., Gaussian or spherical shapes). The degree of freedom for this regression is:

$$dof = N - m \tag{3}$$

where $m$ is the number of parameters defined in the model. For instance, to fit a Gaussian function
to the semivariogram with three parameters ($m=3$), three paired ($N=3$) observations are required at
minimum. It is not feasible to describe $\gamma(h)$ with only one sample. In case of two samples, the



semivariogram might be explained by a line with no offsets (i.e., $\gamma(h) = a_o h$) or a constant
function ($\gamma(h) = b_o$). Different regression models can be used to describe $\gamma(h)$ depending on the
characteristic of the quantity of interest. In this study, we will use a stable Gaussian function:

$$\gamma(h) = a_o(1 - e^{-(\frac{h}{b_0})^{co}}): a_o, b_o, c_o=1.5 \tag{4}$$

A non-linear least-squares algorithm based on Levenberg-Marquardt method will be used to
estimate the regression parameters.
The kriging estimator predicts a value of interest over a defined domain using a
semivariogram model derived from samples [Chilès and Delfiner, 2009]. The kriging model is
defined as [Matheron, 1963]:

$$Z(x) = Y(x) + m(x) \tag{5}$$

where $Y(x)$ is a zero-mean random function, and $m(x)$ is a systematic drift. If we assume
$m(x) = a_o$, the model is called ordinary kriging. Similar to an interpolation problem, the
estimation point ($\hat{Z}(x_0)$), is determined by linearly combining $n$ number of samples, $Z(x_j)$, with
their weights ($\lambda_j$):

$$\hat{Z}(x_0) = \sum_{j=1}^{n} \lambda_j Z(x_j) + \lambda_o \tag{6}$$

where $\lambda_o$ is a constant weight. The mean squared error of this estimation can be written as

$$E(\hat{Z} - Z_o)^2 = \text{Var}\left(\hat{Z} - Z_o\right) + \left[\lambda_o + (\sum_{j=1}^{n} \lambda_j - 1)a_o\right]^2 \tag{7}$$

Where $\hat{Z}$ is the estimation, $Z_o$ is point observations, and $a_o$ is the mean of $Z$ which is unknown. In
order to estimate the weights, we are required to minimize Eq.7, but this cannot be done without
knowing the exact value of $a_o$. A solution is to assume $\lambda_o = 0$ and impose the following condition:

$$\sum_{j=1}^{n} \lambda_j = 1 \tag{8}$$

This condition warrants $E(\hat{Z} - Z_o)$ be zero and removes the need for the knowledge of $a_o$.
Therefore Eq.7 can be written as

$$E(\hat{Z} - Z_o)^2 = \text{Var}\left(\hat{Z} - Z_o\right) = \sum_{j1=1}^{n}\sum_{j2=1}^{n} \lambda_{j1}\lambda_{j2}\gamma_{j1j2} - 2\sum_{j1=1}^{n}\lambda_{j1}\gamma_{j1o} + \gamma_{oo} \tag{9}$$

Using the method of Lagrange multiplier and considering the constraint on the weights, Eq.9 can
be minimized by solving the following problem [Chilès and Delfiner, 2009]:

$$\begin{pmatrix} \lambda_1 \\ \vdots \\ \lambda_n \\ \mu \end{pmatrix} = \begin{pmatrix} \gamma(x_1-x_1) \cdots \gamma(x_1-x_n) 1 \\ \vdots \quad \ddots \quad \vdots \quad \vdots \\ \gamma(x_n-x_1) \cdots \gamma(x_n-x_n) 1 \\ 1 \quad \cdots \quad 1 \quad 0 \end{pmatrix}^{-1} \begin{pmatrix} \gamma(x_1-x_o) \\ \vdots \\ \gamma(x_n-x_o) \\ 1 \end{pmatrix} \tag{10}$$

where $\mu$ is the Lagrange parameter. The first term in the right hand side of this equation shows the
spatial variability described by the semivariogram model among samples, whereas the second term
indicates the modeled variability between samples and the estimation point. The unknowns (the
left hand side of the equation) have a unique solution if, and only if, the semivariogram model is
positive definite and the samples are unique [Chilès and Delfiner, 2009]. The estimation error can
be obtained by



$$\sigma^2 = E(\hat{Z} - Z_o)^2 = \sum_{j=1}^{n} \lambda_j \, \gamma_{jo} - \mu \qquad (11)$$

This equation is an important component in the kriging estimator. Not only can we estimate $Z(x_o)$
given a selection of data points, but also an uncertainty associated with such estimation can be
provided.

### 2.2. Theoretical Cases


*2.2.1. Sensitivity to spatial variability of the field*


The present section illustrates the application of ordinary kriging for several numerical
cases. Five idealized cases are simulated in a grid of 100×100 pixels, namely, a constant field (C1),
a ramp starting from zero in the lower left to higher values in the upper right (C2), an intersection
with concentrated values in four corridors (C3), a Gaussian plume placed in the center (C4), and
multiple Gaussian plumes spread over the entire domain (C5). We randomly sample 200 data
points from each field as is, and successively create the semivariograms in 100 binned distances.
Except C1, which lacks a spatial variability thus $\gamma(h) = b_o = 0$, other semivariograms are fit with
the stable Gaussian function. Using the semivariogram model, we optimize Eq.10 to estimate $\hat{Z}(x)$
for each pixel (i.e., 100×100) with the estimation errors based on Eq.11. Figure 1 depicts the truth
field ($Z(x)$), semivariograms made from the samples, estimated values ($\hat{Z}(x)$), difference of $Z(x)$
and $\hat{Z}(x)$, and error associated with the estimation.
As for C1, the uniformity results in a constant semivariogram leading the estimation to be
identical to the truth. This estimation signifies the unbiased characteristic of ordinary kriging. C1
is never met in reality, however, it is possible to assume some degree of uniformity among data
restrained to background values; a typical example of this can be seen in the spatial distribution of
a number of trace gases in pristine environments such as $NO_2$ [e.g., Wang et al., 2020] and HCHO
[Wolfe et al., 2019]. Under this condition, any data point within the field (i.e., the satellite
footprint) can be assumed to be representative of the spatial variability in truth.
Concerning C2, the semivariogram shows a linear shape meaning data points at larger
distances exhibit larger differences. Generally geophysical samples are uncorrelated at large
distances, thereby one expects the semivarioram to increase more slowly as the distance gets
further. The steady increase in $\gamma(h)$ is indicative of a systematic drift in the data invalidating the
assumption of $m(x) = a_o$. In many applications, a simple polynomial can explain $m(x)$ and
subsequently be subtracted from the data points. An example of this problem is tackled by Onn
and Zebker [2006]; it concerns the spatial variability of water vapor columns measured by GPS
signals. Onn and Zebker [2006] observed a strong relationship between the water vapor columns
and GPS altitudes resulting from the vertical distribution of water vapor in the atmosphere.
Because of this complication, a physical drift model describing the vertical dependency was fit
and removed from the measurements so that they could focus on the horizontal fluctuations. In
terms of C2, one can effortlessly reproduce $Z(x)$ by fitting a three-dimensional plane to barely
three samples, indicating that the semivariogram is of little use.
C3 is an example of an extremely inhomogeneous field manifested in the stabilized
semivariogram at a value of $\gamma$ (~500), called the sill, indicating insignificant information (variance)
from the samples beyond this distance (~20), called the range. Range is defined as the separation
distance at which the total variance in data is extracted. The smaller the range is, the more
heterogeneous the samples will be. While the estimated field roughly captures the shape of the
intersections, it is spatially distorted at places with relatively sparse data points. The kriging model



error is essentially a measure of the density of information. It converges to zero in the sample's
location and diverges to large values in gaps.
C4 is a close example of a point source emitter with faint winds and turbulence. The
semivariogram exhibits a bell shape. As samples get further from the source, the variance diverges,
stabilizes, and then sharply decreases. This is essentially because many data points with low
values, apart from each other, have negligible differences. This tendency is recognized as the hole
effect which is characterized for high values to be systemically surrounded by low values (and
vice versa). It is possible to mask this effect by fitting a semivariogram model stabilizing at certain
sill (like the one in Figure 1). Nonetheless, if the semivariogram shows periodic holes, the fitted
model should be modified to a periodic cosine model [Pyrcz and Deutsch, 2003].
The last case, C5, shows a less severe case of the hole effect previously observed in C4.
This is due to the presence of more structured patterns in different parts of the domain. The range
is roughly twice as large as the previous case (C4) denoting that there is more information
(variance) among the samples at larger distances. A number of experiments using this particular
case will be discussed in the following subsections.
*2.2.2. Sensitivity to the number of samples*
It is often essential to optimize the number of samples used for kriging. The kriging
estimator somewhat recognizes its own capability at capturing the spatial variability through
Eq.11. Thus, if the target phenomenon is spatially too complex and/or the samples are too limited,
the estimator essentially informs that $\hat{Z}(x_0)$ is unreliable through large variance. However, there
is a caveat; $Y(x)$ must be a Gaussian random model with a zero mean so that kriging can capture
the statistical distribution of $\hat{Z}$ given the data points. Except this case, the kriging variance can
either be underestimated or overestimated depending on the level of skewness of the statistical
distribution of $Y(x)$ [Armstrong, 1994]. Figure 2 shows the kriging estimation for C5 using 5, 25,
50, 100, and 500 random samples in the entire field. Immediately apparent is a better description
of the semivariogram when larger number of samples are used, which in turn, results in a better
estimation of $Z(x)$. The optimum number of samples to reproduce $Z(x)$ depends on the
requirement for the relative error ($\sigma/Z(x)$) being met at a given location.
*2.2.3. Sensitivity to the tessellation of samples*
A common application of kriging is to optimize the tessellation of data points for a fixed
number of samples to achieve a desired precision. In real-world practices, the objective of such
optimization is very purpose-specific, for example, one might prefer a spatial model representing
a certain plume in the entire domain. Different ways for data selection exist [e.g., Rennen, 2008],
but for simplicity, we focus on four categories: purely random, stratified random, a uniform grid,
and an optimized tessellation. Figure 3 demonstrates the estimation of C5 using 25 samples chosen
based on those four procedures.
Concerning the random selection, the lack of samples over two minor plumes cause the
estimation to deviate largely from the truth. While a random selection may seem to be practical
because it is independent of the underlying spatial variability, it can suffer from under sampling
issues, thus being inefficient. As a remedy, it might be advantageous to group the domain into
similar zones. We classify the domain into four zones using the k-mean algorithm (not shown) and
randomly sample six to seven points from each one (total 25). We achieve a better agreement
between the estimated field and the truth because we exploited some prior knowledge (here the
contrast between low and high values).
As for the uniform grid, we notice that there are fewer data points in the semivariogram
stemming from redundant distances which is indicative of correlated information. Nonetheless, if



the desired tessellation is neutral with regard to location meaning that all parts of the domain is
equal of scientific interest, the uniform grid is the most optimal design for the prediction of $Z(x)$
under an ideally isotropic case. A mathematical proof for this claim can be found in Chilès and
Delfiner [2009].
To execute the last experiment, we select 25 random samples for 1000 times and find the
optimal estimation by finding the minimum sum of $|\hat{Z}(x_0) - Z(x)|$. It is worth mentioning that
the optimized tessellation is essentially a local minimum based on 1000 realizations. The
optimized location of samples seems to more clustered over areas with large spatial gradients. Not
too surprisingly, we observe the smallest discrepancy between the estimation and the truth.
A lingering concern over the application of these numerical experiments is that the truth is
assumed to be known. The truth is never known, by this means we may never exactly know how
well or poorly the kriging estimator is performing. However, it is highly unlikely for some prior
understandings or expectations of the truth to be absent. If this is the case, which is rare, a uniform
grid should be intuitively preferred to deliver the local estimations of average values in uniform
blocks. In contrast, if the prior knowledge is articulated by previous site visits, model predictions,
theoretical experiments, pseudo-observations, or other relevant data, the tessellation needs to be
optimized.
It is important to recognize that the uncertainties associated with the prior knowledge
directly affects the level of confidence in the final answer. Accordingly, the prior knowledge error
should ultimately be propagated to the kriging variance. The determination of the prior error is
often done pragmatically. For example, if the goal is to design the location of thermometer sites to
capture surface temperature during heat waves using a yearly averaged map of surface
temperature, it would be wise to specify a large error with this specific prior information to play
down the proposed design. This is primarily because the averaged map underrepresents such an
atypical case. A possible extension of this example would be to use a weather forecast model with
quantified errors capable of capturing retrospective heat waves. Although a reasonable forecast in
the past does not necessarily guarantee a reasonable one in the future, it is rational to assume for
the uncertainty with a new tessellation design using the weather model forecast to be lower than
that of using the averaged map.
A general roadmap for the data tessellation design is shown in Figure 4. As proven in Chilès
and Delfiner [2009], if the field is purely isotropic, the uniform grid is the most intuitive sensible
choice when the prior information on the spatial variability is lacking. When the prior knowledge
with quantified errors is available, an optimum tessellation can be achieved by running a large
number of kriging models with suitable $\gamma(h)$ and picking the one yielding the minimum distance
between the prior knowledge and the estimation. The choice of the cost function (here L1 norm)
is purpose-specific. For example, if the reconstruction of a major plume was the goal, using a
weighted cost function, geared towards capturing the shape of plume, would be more appropriate.
*2.2.4. Sensitivity to the grid size*
A kriging model can estimate a geophysical quantity at a desired location considering the
data-driven spatial variability information. Since the kriging model is practically in a continuous
form, the desired locations can be anywhere within the field of $V$. A question is whether or not it
is necessary to map the data onto a very fine grid. There is a trade-off between the computational
cost and the accuracy of the interpolated map. The range of the underlying semivariogram helps
in finding the optimal solution. The greater the range (i.e., a more homogeneous field), the less
important to map the data in a finer grid.



Figure 5a depicts an experiment comparing the estimates of C2 at different grid sizes with
the truth. The departure of the estimate from the truth is rather negligible for several coarse grids
(e.g., 10×10). The homogeneous field, manifested by the large range (Figure 1), allows for a
reasonable estimation of $Z(x)$ at coarse resolutions with inexpensive computational costs. Figure
5b shows the same experiment but on C5 with the optimized tessellation. As opposed to the
previous experiment, the estimate substantially diverges from the truth when increasing the grid
size, suggesting that a finer resolution should be used for fields with smaller ranges (i.e.,
heterogeneous fields).
The complexity of directly using the range for choosing the optimal grid cell size arises
from the fact that the level of spatial homogeneity can vary within the domain. In fact, the range
is derived from a semivariogram model representing a crude estimate of varying ranges occurring
at various scales. It is intuitively clear that depending on the degree of heterogeneity, which is
spatiotemporally variable, the grid size needs to be adaptively adjusted [Bryan, 1999]. For the sake
of simplicity, but at a higher computational cost, we adopt a numerical solution which is to first
simulate on a coarse grid, then on a finer one until the difference with respect to the previous grid
size across all pixels reaches to an acceptable value (<1%). We name this output (1×1) with the
optimized tessellation for C5 as C5opt.

**3. Comparison of points to satellite pixels**

*3.1. Synching the scales between the gridded field and satellite pixels*

To minimize the complications of different spatial scales between two gridded data, we
first need to upscale the finer resolution data to match the coarse ones. In case of numerical
chemical transport or weather forecast models, the size of the grid is definitive. Likewise, a satellite
footprint, mainly dictated by the sensor design, the geometry, and signal-to-noise requirements
[Platt et al., 2021], is known. However, the grid size of the kriging estimation is a variable subject
to optimization which has been discussed previously.
When we compare the grid size of the kriging estimate to that of a satellite (or a model),
three situations arise: First, the kriging spatial resolution is coarser than the satellite, a condition
occurring when either the field is homogeneous or the field is under sampled. In situations where
the field is homogeneous ($\gamma(h) \cong 0$), it is safe to directly compare the data points to the satellite
measurements without having to use kriging. If the under sampling is the case (see Figure 2 with
5 samples), it is sensible to first investigate if the field is homogeneous within the satellite footprint
using different data (if any). If the homogeneity is met, we either can compare two datasets without
kriging or to match the size of kriging grid cell with the satellite footprint and statistically involve
the kriging variance in the comparison (discussed later); nonetheless, the kriging estimate beyond
the location of samples must be used with extra caution because their variance very quickly
departures from zero to extremely large numbers (see Figure 1). Thus, there is a compromise
between increasing the number of paired samples between two datasets and enhancing the level of
confidence in statistics. If independent observations suggest that there might be large heterogeneity
within a satellite footprint, it is strongly advised against quantitatively comparing the points to the
satellite observations. Second, the number of samples is fewer than three observations in the field
so it is in principal impossible to build a semivariogram. Validating a satellite under this condition
is prone to misinterpretation because the spatial heterogeneity cannot be modeled. Nonetheless, if
one presumes a good degree of homogeneity within the sensor footprint (such as very high-
resolution remote sensing airborne data), the direct comparison of point measurements might be
possible. Third, the satellite footprint is coarser than the kriging estimate. Under this condition, we
upscale the kriging map to match the spatial resolution of the satellite using





$$\hat{Z}_c = \hat{Z}_f * S = \int \hat{Z}_f(x) S(x - y) dy \tag{12}$$

where $S$ is the spatial response function, $\hat{Z}_c$ is the coarse kriging field, <*> is the convolution
operator, $y$ is shift, and $\hat{Z}_f$ is the fine field. In discrete form we can rewrite Eq.12 in

$$\hat{Z}_c[i,j] = \sum_m \sum_n \hat{Z}_f[i-m, j-n] \, S[m,n] \tag{13}$$

where $m$ and $n$ are the dimension of the response function. The mathematical formulation of
$S[m,n]$ for a number of satellites can be represented by two-dimensional super Gaussian functions
as discussed in Sun et al. [2018]. Atmospheric models have a uniform response to the simulated
values within a grid, therefore $S[m,n] = \frac{1}{m \times n} J_{m,n}$, where $J$ is the matrix of ones. In the same way,
the kriging variance should be convolved through

$$\sigma_c^2[i,j] = \sum_m \sum_n \sigma_f^2[i-m, j-n] \, S^2[m,n] \tag{14}$$

where $\sigma_c^2$ and $\sigma_f^2$ are the kriging variance in the coarse and the fine grids, respectively.
To demonstrate the upscaling procedure, we use C5opt (1×1) and upscale it at six grids
$(m,m)$ of 5×5, 10×10, 15×15, 20×20, 25×25, and 30×30 considering $S = \frac{1}{m^2} J_{m,m}$. Figure 6 shows
the resultant map overplotted with the samples along with the error estimation. Two tendencies
from this experiment can be identified: First, the discrepancy of the point data and $\hat{Z}$ is becoming
more noticeable as the grid size grows; this directly speaks to the notion of the spatial
representativeness; large grid cells are less representative of sub-grid values. Second, the gradients
of the field along with the estimation error become smoother primarily due to convolving the field
with the spatial response function, which acts as a low pass filter.
We further directly compare $\hat{Z}$ to the samples (i.e., observations) shown in Figure 7. We
see an excellent comparison between $\hat{Z}$ at 1×1 resolution with the observations underscoring the
unbiasedness characteristic of the kriging estimator. Conversely, the upscaled field gradually
diverges from the observations. This divergence is *the problem of scale*.

### 3.2. Point to pixel vs pixel to pixel

To elaborate on the problem of scale, we design an idealized experiment theoretically
validating pseudo satellite observations against some pseudo point measurements. The pseudo
satellite observations are created by upscaling the C5 truth ($Z$) to 30×30 grid footprint considering
$S = \frac{1}{m^2} J_{m,m}$, meaning that the satellite is observing the truth but in a different scale (not shown).
The pseudo point measurements are the ones used for C5opt. Figure 8a shows the direct
comparison of the satellite pixel with the point observations. By ignoring the fundamental fact that
these two datasets are inherently different in nature, displaying the same geophysical quantity by
at different scales, we observe a perceived discrepancy ($r^2=0.64$). The comparison suggests a
wrong conclusion that the satellite observations are biased-low. This discrepancy is unrelated to
any observational or physical errors, rendering any physical interpretation of the comparison
biased due to spatial-scale differences in the data sets. Figure 8b depicts the comparison of each
grid of the upscaled kriging estimate (30×30) with that of the satellite. This direct comparison
shows a strong degree of agreement ($r^2=0.98$), shaking off the erroneous idea of directly comparing
point to gridded data when the field exhibits substantial spatial heterogeneity.



Yet, the comparison misses an important point: the kriging estimate is considered error-
free. We attempt to incorporate the kriging variance through a Monte Carlo linear regression
method. Here, the goal is to find an optimal linear fit ($y = ax + b + \varepsilon$) such that $\chi^2 =$
$\sum \frac{[y - f(x_i, a, b)]^2}{\sigma_y^2 + a^2 \sigma_x^2}$ is minimized. $\sigma_y^2$ and $\sigma_x^2$ are the variances of $y$ (here the satellite) and $x$ (the kriging
variance), respectively. We set the errors of $y$ to zero, and randomly perturb the errors of $x$ based
on a normal distribution with zero mean and a standard deviation equal to that of kriging estimate
15,000 times. The average of optimized $a$ and $b$ coefficients derived from each fit are then
estimated and their deviation at 95% confidence interval assuming a Gaussian distribution is
determined. Figure 8b,c show the linear fit with and without considering the kriging error estimate.
The linear fit without involving the kriging error gives a strong impression that it is nearly perfect,
following closely to the paired observations. This is essentially explainable by the primary goal of
$\chi^2$ which is to minimize the L2 norm of residuals ($y - f(x_i, a, b)$), portraying a very optimistic
picture of the satellite validation. The linear fit considering the kriging errors is different. The
uncertainties associated with $a$ and $b$ are larger since $x$ is variable (shown in horizontal error bars).
The optimal fit gravitates towards the points with smaller standard deviations as they possess a
larger weight. The confidence in the linear fit at higher values is lower due to their errors being
large. This fit is a more realistic portrayal of the satellite validation.
Figure 9 summarizes the general roadmap for satellite validations against point
measurements. To fit the semivariogram with at least two parameters, we are required to have
three samples at minimum. Therefore, it is implausible to derive the spatial information from the
point data where sampling is extremely sparse (<3 samples within the field). The only case of
directly comparing point and satellite pixels is when the field within satellite footprint or the field
in general is rather homogeneous confirmed by independent data/models. Having more samples
allows to acquire some information on the spatial heterogeneity. The information carried by the
data is considered more and more robust with increasing the number of samples. Subsequently,
the kriging map along with its variance derived from a reasonable semivariogram at an optimized
grid resolution should be convolved with the satellite response function so that we can conduct an
apples-to-apples comparison. A real-world example on the satellite validation will be shown later.
**4. Real-world experiments**
*4.1. Spatial distribution of NO$_2$*

We begin with focusing on tropospheric NO$_2$ columns observed by TROPOMI sensor
[Copernicus Sentinel data processed by ESA and Koninklijk Nederlands Meteorologisch Instituut
(KNMI), 2019; Boersma et al., 2018] at ~13:30 LST. We choose NO$_2$ primarily due to its spatial
heterogeneity [e.g., Souri et al., 2018; Nowlan et al., 2016, 2018; Valin et al., 2011; Judd et al.,
2020]. We oversample good quality pixels (qa_flag>0.75) through a physical-based gridding
approach [Sun et al., 2018] over Texas at 3×3 km$^2$ resolution in four seasons in 2019. We extract
samples by uniformly selecting the NO$_2$ columns in the center of each 30×30 km$^2$ block. The
semivariogram along with its model are calculated, and then we krige the samples. Figure 10 shows
the NO$_2$ columns map for four different seasons, the semivariogram, the kriging estimates, and the
differences between the estimate and the field. High levels of NO$_2$ are confined to cities indicating
the sources being predominantly anthropogenic. Wintertime NO$_2$ columns are larger than
summertime mainly due to meteorological conditions and the OH cycle, the major sink of NO$_2$.
All semivariograms exhibit the hole effect. This is because of high values of NO$_2$ being
systematically surrounded by low values. Regardless of the season, we fit the stable Gaussian to
variances at distances smaller than 2.5° (~275 km$^2$). The $b_0$ parameter explaining the range (or the



length scale) is found to be 0.94, 0.88, 0.71, and 0.83 degree for DJF, MAM, JJA, and SON,
respectively. These numbers strongly coincide with the length scale of $NO_2$; wintertime $NO_2$
columns are spatially more uniform around the sources thus in relative sense, they are more
homogeneous (spatially correlated) than those in warmer seasons. On the other hand, the shorter
$NO_x$ lifetime in summer results in a steeper gradient of $NO_2$ concentrations. This tendency should
not be generalized because transport and various $NO_x$ sources including biomass burning, soil
emissions, and lightning and can have large spatiotemporal variability resulting in different length
scales in different times of a year. The differences between the kriging estimate and the field show
some spatial structures indicating that $NO_2$ is greatly heterogenous.

### 4.2. Optimized tessellation over Houston

The preceding TROPOMI data enabled us to optimize a tessellation of ground-based point
spectrometers over Houston. Our goal here is to propose an optimized network for winter 2021
given our knowledge on the spatial distribution of $NO_2$ columns in winter 2019 measured by
TROPOMI. The assumption of using a retrospective $NO_2$ field for informing a hypothetical future
campaign is not entirely unrealistic. If we have a consistent number of pixels from TROPOMI
between two years, it is unlikely for the spatial variance of $NO_2$ to be substantially different for
the same season. We follow the framework proposed in Sect. 2.2.3 involving randomly selecting
samples from the field (for 50000 iteration), and calculating kriging estimates for a given number
of spectrometers. We then chose the optimum tessellation based on the minimum sum of $|\hat{Z}(x_0) -
Z(x)|$.
Figure 11 shows the optimized tessellation given 5, 10, 15, and 20 spectrometers over
Houston. The Houston plume is better represented with more samples being used. All cases share
the same feature; the optimized samples are clustered in the proximity or within the plume. This
tendency is clearly intuitive. We are required to place the spectrometers in locations where a
substantial gradient (variance) in the field is expected. The kriging estimate using 20 samples does
not substantially differ in comparison to the one using 15 samples. A preferable strategy is to keep
the number of spectrometers as low as possible while achieving a reasonable precision. Based on
the presented results, the optimized tessellation using 15 samples is preferred among others.

### 4.3. Validating OMI tropospheric $NO_2$ columns during DISCOVER-AQ 2013 campaign using Pandora

In order to understand ozone pollution [e.g., Mazzuca et al., 2016; Pan et al., 2017; Pan et
al., 2015], characterize anthropogenic emissions [Souri et al., 2016, 2018], and validate satellite
data [Choi et al., 2020], an intensive air quality campaign was made in September 2013 over
Houston (DISCOVER-AQ). The campaign encompassed a large suite of Pandora spectrometer
instrument (PSI) (11 stations) measuring total $NO_2$ columns with a high precision ($2.7 \times 10^{14}$
molecules cm$^{-2}$) and a moderate nominal accuracy ($2.7 \times 10^{15}$ molecules cm$^{-2}$) under the clear-sky
condition [Herman et al., 2007]. We remove the observations with an error of >0.05 DU,
contaminated by clouds, and averaged them over the month of September at 13:30 LST (± 30
mins). We attempt to validate OMI tropospheric $NO_2$ columns version 3.0 [Bucsela et al., 2013]
refined in Souri et al. [2016] with the 4-km model profiles. The OMI sensor resolution varies from
$13 \times 34$ km$^2$ at nadir to ~$40 \times 160$ km$^2$ at the edge of the scan line. Biased pixels were removed based
on cloud fraction > 0.2, terrain reflectivity > 0.3, and main (xtrack) quality flags =0. Following
Sun et al. [2018], we oversample high quality pixels in the month of September 2013 over Houston
at 0.2° resolution. To remove the stratospheric contributions from PSI measurements, we subtract
their columns from those of OMI stratospheric $NO_2$ over the area ($2.8 \pm 0.16 \times 10^{15}$ molecules cm$^{-2}$).
Figure 12 shows the monthly-averaged tropospheric $NO_2$ columns measured by OMI overplotted



by 11 PSIs. The elevated $NO_2$ levels (up to ~6×10$^{15}$ molecules cm$^{-2}$) are seen over the center of
Houston.
We then follow the validation framework shown in Figure 9 in which the number of point
measurements and the level of heterogeneity are the main factors in deciding if we should directly
compare them to the satellite pixels. Figure 13 shows the monthly-averaged PSI measurements
along with the semivariogram and resulting kriging estimate at an optimized resolution (~2 km$^2$ =
13800 data over the entire region) and errors. The distribution of semivariogram suggests that there
is a strong degree of spatial heterogeneity, necessitating the use of kriging. We fit a stable Gaussian
to the semivariogram resulting in $2.23 \times (1 - e^{-(\frac{h}{0.19})^{1.5}})$. The spatial information (variance) levels
off at 0.19° (~21 km) with a maximum variance equal to 2.23 molecules$^2$ cm$^{-4}$. The measurements
beyond this range (21 km) have a minimal weight due to this length scale. It is because of this
reason that we see the kriging estimate converges to a fixed value at the grids being further than
this range. The kriging errors of those grid cells are constantly large (40% relative error). The
optimum grid size for kriging is found to be 2 km$^2$ (<1% difference across all grids). Subsequently,
we use the super Gaussian spatial response function described in Sun et al. [2018] to convolve
both the kriging estimate and error within. Figure 14 shows the differences between the kriging
estimate and error before and after convolution. The response function (OMI pixel) tends to be on
average coarser than 2 km$^2$ resulting in smoothing of both the kriging estimate and error.
We ultimately conduct two different sets of comparison: directly comparing PSI to OMI
pixels, and comparing convolved kriged PSI to OMI. It is worth noting that PSI measurements are
monthly-averaged; similarly OMI data are oversampled in a monthly basis. In terms of the PSI,
we only account for grid cells whose kriging error is below 1.2×10$^{15}$ molecules cm$^{-2}$ (1193
samples, 8% of total kriging grids). As for the grid to grid comparison, the kriging variance is
considered in the linear polynomial fitted to the data through the Monte Carlo of chi-square
minimization with 5,000 iterations. The variability with the OMI stratospheric $NO_2$ columns (0.16
×10$^{15}$ molecules cm$^{-2}$) is added to the PSI error for both analyses. The left and right panels of
Figure 15 show the comparisons. As for the direct comparison of actual points (PSI) to pixels
(OMI), the PSI measurements indicate a deviation of the slope (r$^2$=0.66) from the unity line. This
suggests that there is an unresolved magnitude-dependent systematic error. The grid-to-grid
comparison not only offers a clearer picture of the distribution of data points, but also it hints at
the offset being rather constant (0.66±0.18×10$^{15}$ molecules cm$^{-2}$; r$^2$=0.72). We also observe that
the statistics between the satellite and the benchmark are moderately improved. This comparison
in general provides an important implication: the varying offsets in a plume shape environment
(high to low values) are not necessarily due to variable offsets in the satellite retrieval, as the
kriging estimate suggests that those varying offsets in point-to-point comparison, manifested in
slope = 0.76, are a result of varying spatial scales.
**Summary**
There needs to be increased attention to the spatial representativity in the validation of
satellite (model) against pointwise measurements. A point is the element of space, whereas satellite
(model) pixels (grids) are (at best) the product of the integration of infinitesimal points and a
normalized spatial response function. If the spatial response function is assumed to be an ideal
box, the resulting grid will represent the average. Essentially, no justifiable theory exists to accept
that the averaged value of a population should absolutely match with a sample, unless all samples
are identical (i.e., a spatially homogeneous field). This glaring fact is often overlooked in the
atmospheric science community. At a conceptual level, we are required to translate pointwise data
to grid format (i.e., rasterization). This can be done by modeling the spatial autocorrelation (or



semivariogram) extracted from the spatial variance (information) among measured sample points.
Assuming that the underlying field is a random function with an unknown mean, the best linear
unbiased predictions of the field can be achieved by kriging using the modeled semivariograms.
In this study, we discussed methods for the kriging estimation of several idealized cases.
Several key tendencies were observed through this experiment: first, the range corresponded to the
degree of spatial heterogeneity; a larger range indicated the less presence of heterogeneity. Second,
the kriging variance explaining the density of information quickly diverged from zero to large
values when the field exhibited large spatial heterogeneity. This tendency mandates increasing the
number of samples (observations) for those cases. Third, while the semivariogram models were
constructed from discrete pair of samples, they are mathematically in a continuous form. It is
because of this reason that we determined the optimal spatial resolution of the kriging estimate by
incrementally making the grids finer and finer until a desired precision was met.
The present study applied kriging to achieve an optimum tessellation given a certain
number of samples such that the difference between our prior knowledge of the field, articulated
by previous observations, models or theory, and the estimation is minimal. Usually there is
uncertainty about the prior knowledge that should be propagated to the final estimates. The
optimum tessellation for a range of idealized and real-world data consistently voted for placing
more samples in areas where the gradients in the measurements were significant such as those
close to point emitters.
This study also revisited the spatial representativity issue; it limits the realistic
determination of biases associated with satellites (models). In one experiment, we convolved the
kriging estimate for a multi-plume field with a box filter but various sizes. The perfect agreement
($r=1.0$) between the samples (point) and kriging output (pixel) seen at a high spatial resolution
gradually vanished with coarsening of the grids ($r=0.8$). We also directly compared samples (point)
with pseudo satellite observations (showing the truth) with a coarse spatial resolution which led to
a flawed conclusion about the satellite being biased-low. We modeled the semivariogram of those
samples, estimated the field using kriging, and convolved with the pseudo-satellite spatial response
function. The direct comparison of this output with that of the satellite showed a completely
different story suggesting that the data were rather free of any bias. A serious caveat with using a
spatial model (here kriging) is that it consists of errors: the estimations being further from samples
are less certain. It is widely known that discounting the measurement/model errors in true straight-
line relationship between data can introduce artifacts. To consider the kriging variance in the
comparisons we employed a Monte Carlo method on chi-square optimization which ultimately
allowed us to not only provide a set of solutions within the range of the uncertainty of the kriging
model, but also to assign smaller weights on gross estimates.
We further validated monthly-averaged Ozone Monitoring Instrument (OMI) tropospheric
$NO_2$ columns using 11 Pandora Spectrometer Instrument (PSI) observations over Houston during
NASA's DISCOVER-AQ campaign. A pixel-to-point comparison between two dataset suggested
varying biases in OMI manifested in a slope far from the identity line. By contrast, the kriging
estimate from the PSI measurements, convolved with the OMI spatial response function, resulted
in an inter-comparison slope close to the unity line. This suggested that there was only a constant
systematic bias ($0.66\pm0.18\times10^{15}$ molecules cm$^{-2}$) associated with the OMI observations which
does not vary with increasing tropospheric $NO_2$ column magnitudes.
The central tenants of satellite and model validation are pointwise measurements. Our
experiments paved the way for a clear roadmap explaining how to transform these pointwise





datasets to a comparable spatial scale relative to satellite observations. It is no longer necessary to ignore *the problem of scale*. The comparisons can be carefully conducted in the following steps:

i. Construct the experimental semivariogram if the number of point measurements allows (usually >= 3 within the field; the field can vary depending on the length scale of the compound).

ii. Drop the quantitative assessment if the number of point measurements are insufficient to gain spatial variance and the prior knowledge suggests a high likelihood of spatial heterogeneity within the field.

iii. Choose an appropriate function to model the semivariogram.

iv. Estimate the field with kriging (or any other spatial estimator capable of digesting the semivariogram) and calculate the variance.

v. Estimate the optimum grid resolution of the estimate.

vi. Convolve the kriging estimate and its variance with the satellite (model) spatial response function (which is sensor specific).

vii. Conduct the direct comparison of the convolved kriged output and the satellite (model) considering their errors through a Monte Carlo (or at minimum a weighted least-squared method).

Recent advances in satellite trace gas retrievals and atmospheric models have helped extend our understanding of atmospheric chemistry but an important task before us in improving our knowledge on atmospheric composition is to embrace the semivariogram (or spatial auto-correlation) notion when it comes to the point-pixel comparisons, so that we can have more robust quantitative applications of the data and models.

**Acknowledgement**
Amir Souri and Matthew Johnson were funded for this work through NASA's Aura Science Team (grant number: 80NSSC21K1333). Kang Sun acknowledges support by NASA's Atmospheric Composition: Modeling and Analysis (ACMAP) program (grant number: 80NSSC19K09). We thank many scientists whose concerns motivated us to tackle the presented problem. In particular, we thank Chris Chan Miller, Ron Cohen, Jeffrey Geddes, Gonzalo González Abad, Christian Hogrefe, Lukas Valin, and Huiqun (Helen) Wang.

**Author contributions**
AHS designed the research, executed the experiments, analyzed the data, made all figures, and wrote the paper. KS implemented the oversampling method, provided the spatial response functions, and oversampled TROPOMI data. KC, XL, and MSJ helped with the conceptualization of the study and the interpretation of the results. All authors contributed to discussions and edited the paper.



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



Figures:

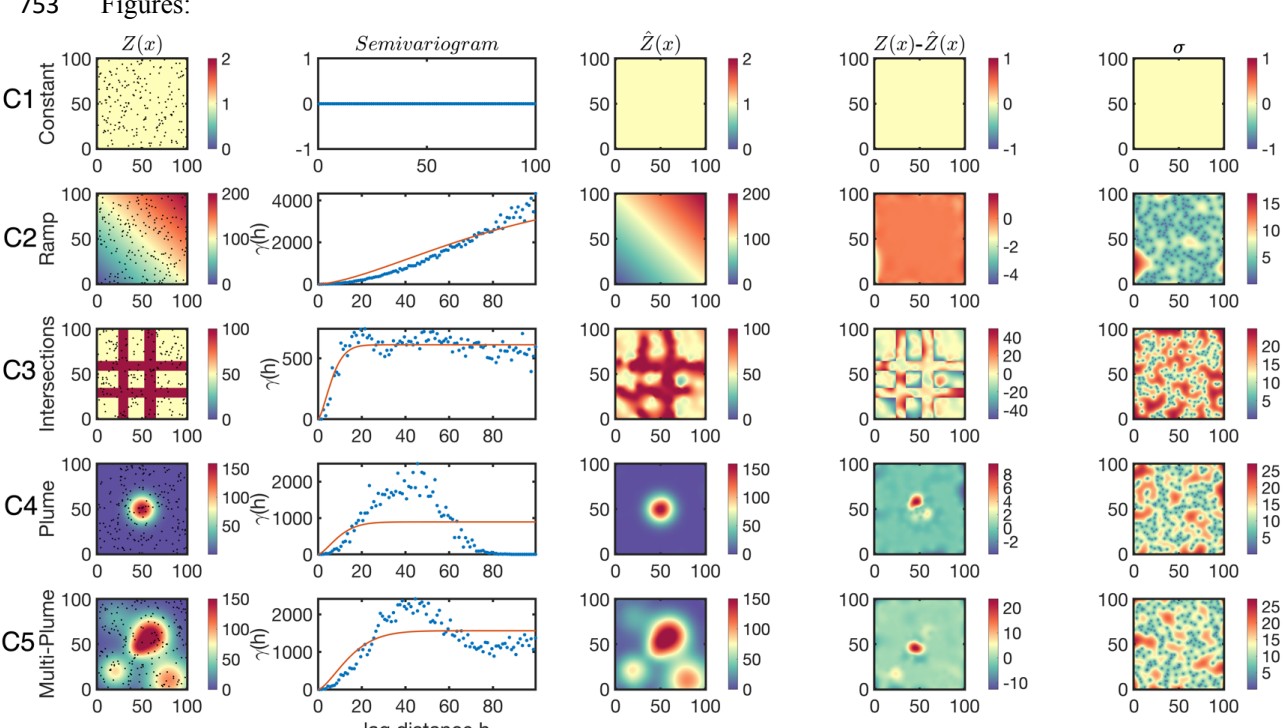

**Figure 1**. (first column) Five theoretical fields randomly sampled with 200 points (dots), namely,
a constant field (C1), a ramp starting from zero in the lower left to higher values in the upper right
(C2), an intersection with concentrated values in four corridors (C3), a Gaussian plume placed in
the center (C4), and multiple Gaussian plumes spread over the entire domain (C5). (second column)
the corresponding isotropic semivariograms computed based on Eq.2; the red line shows the stable
Gaussian fitted to the semivariogram based on Levenberg-Marquardt method. (third column) The
kriging estimate at the same resolution of the truth (i.e., 1×1) based on Eq.6. (fourth column) The
difference between the estimate and the truth. (fifth column) the kriging standard error based on
Eq.11.




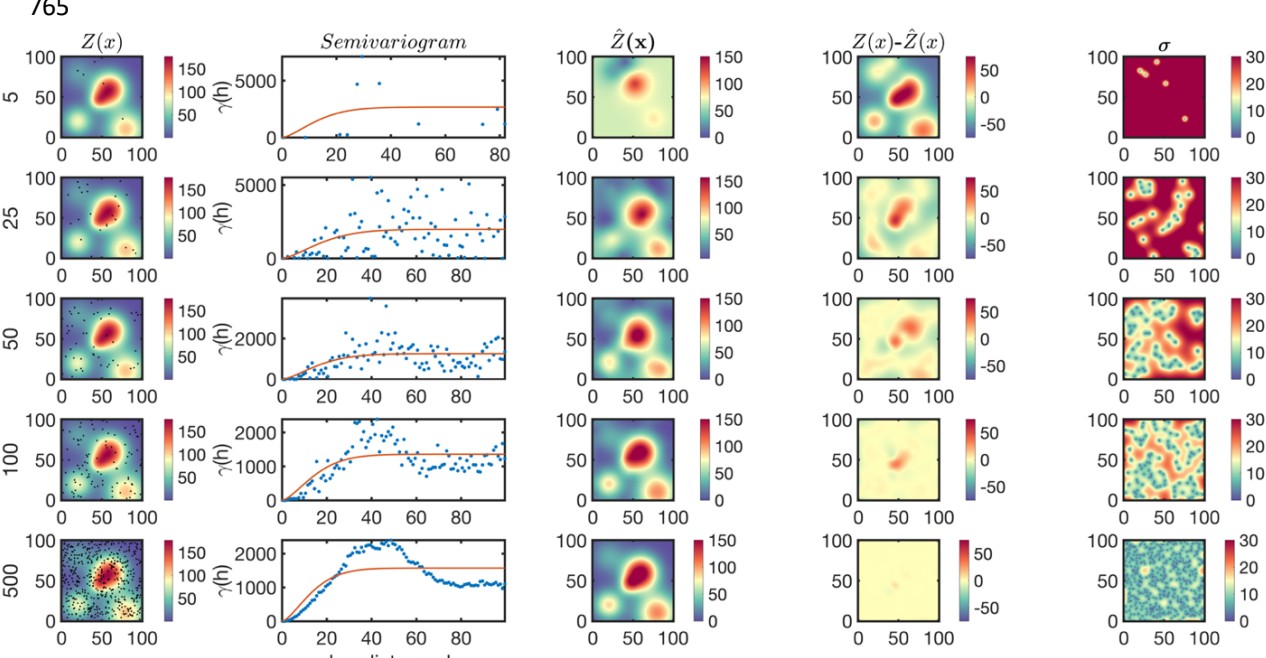

**Figure 2.** (first column) The multi-plume case (C5) randomly sampled with different number of samples (5, 25, 50, 100, and 500), (second column) the corresponding isotropic semivariogram, (third column) the kriging estimate, (fourth column) the difference between the estimate and the truth, and (fifth column) the kriging standard error.


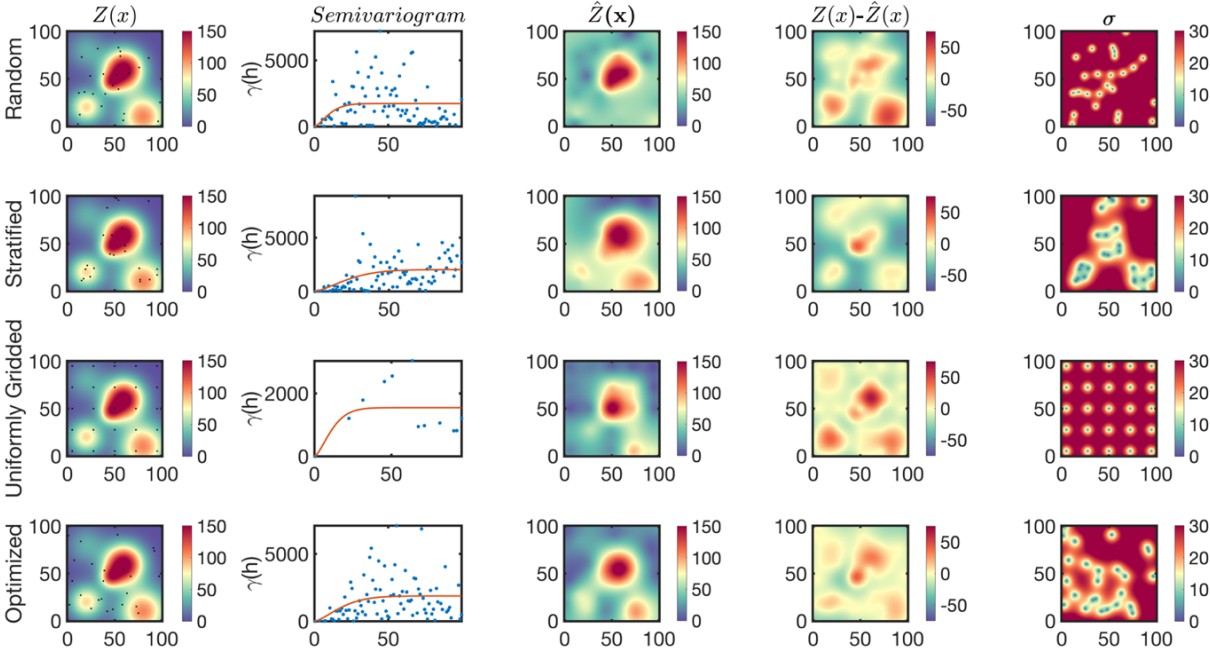

**Figure 3.** The multi-plume case (C5) randomly sampled by four different sampling strategies using a constant number of samples (25). The sampling strategies include purely random (first row), stratified random (second row), uniform grids (third row), and an optimized tessellation proposed based on kriging (fourth row). Columns represent the truth, the isotropic semivariogram, the kriging estimate, the difference between the estimate and the truth, and the kriging standard error.

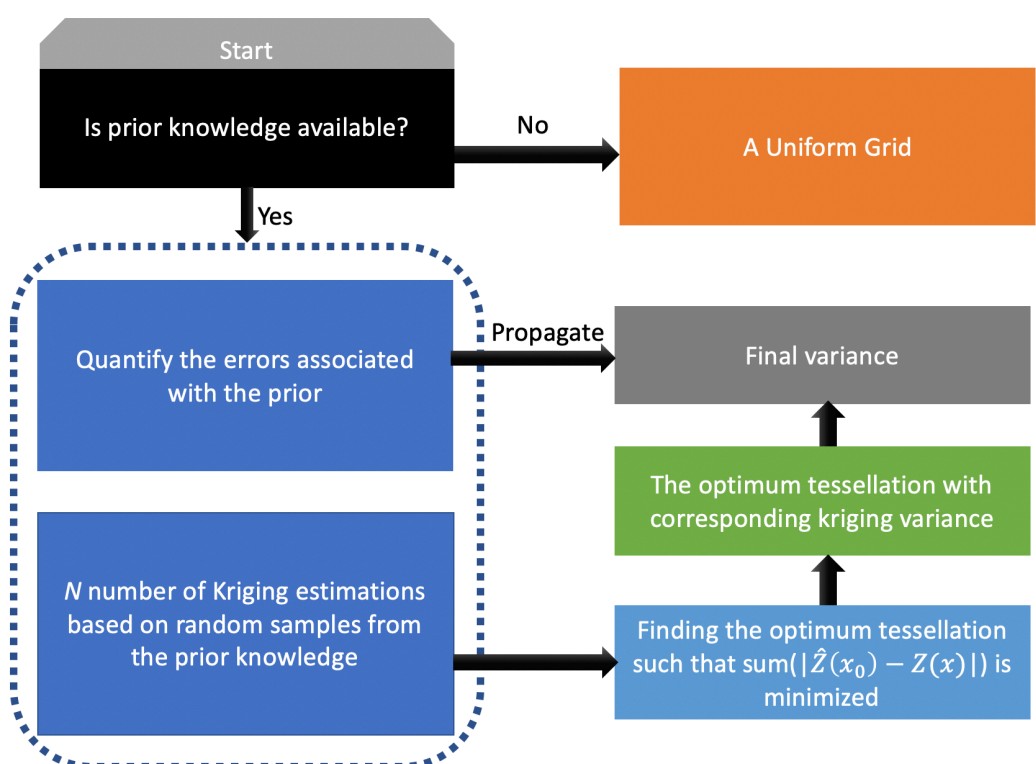

**Figure 4.** A schematic illustrating a framework for optimum sampling (tessellation) strategy. The
prior knowledge refers to any data being able of describing our quantity of interest including site-
visits, theoretical models, satellite observations, emissions, and etc.



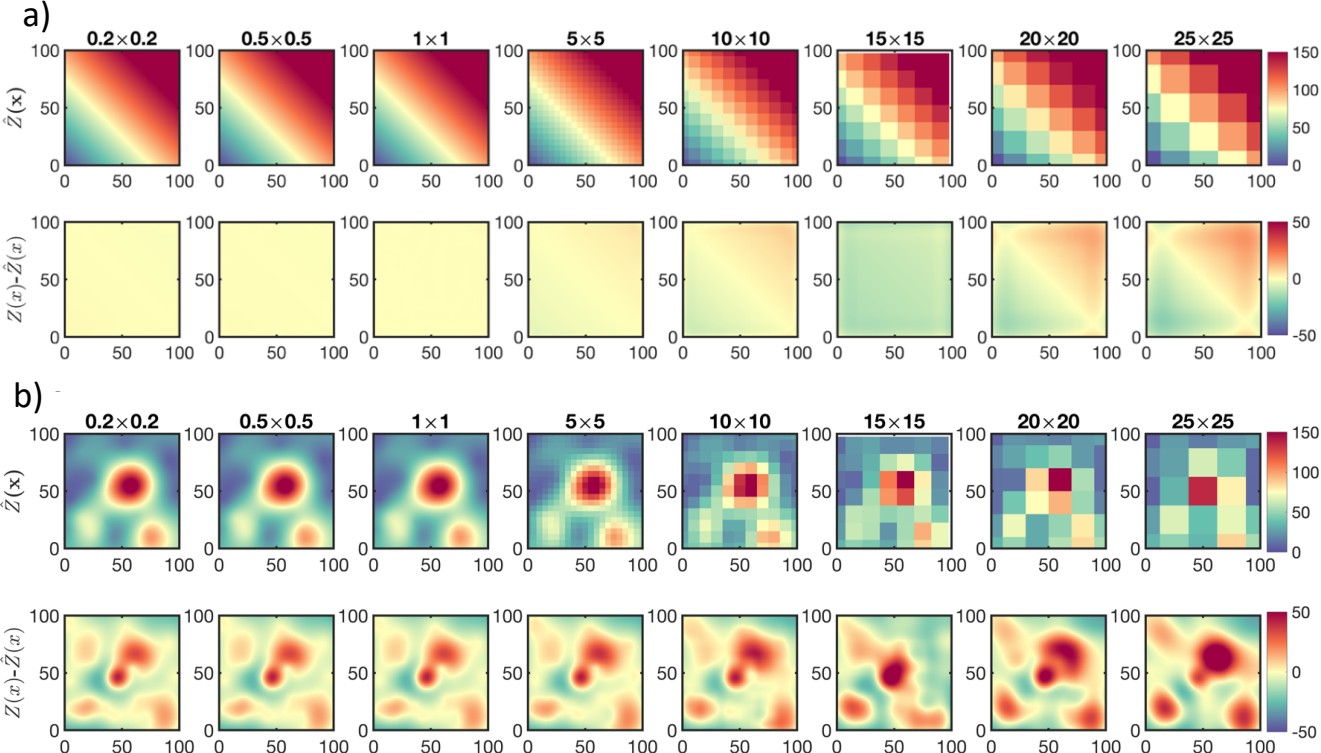

**Figure 5.** Finding an optimum grid cell for kriging. (a) The kriging estimates of the ramp (C2) at different grid resolutions ranging from 25×25 pixel to 0.2×0.2. (b) The kriging estimates of the multi-plume (C5) with optimized samples shown in Figure 3 for different grid resolutions. C2 is more homogeneous than C5, as a result, it is less sensitive to the resolution of the kriging estimate. The optimum grid resolution for C2 is 10×10, whereas it is 1×1 for C5. These numbers are based on observing negligible difference (<1%) between the kriging estimate at the optimum resolution and the one computed at a finer resolution step. We call the optimum output for C5 as C5opt.





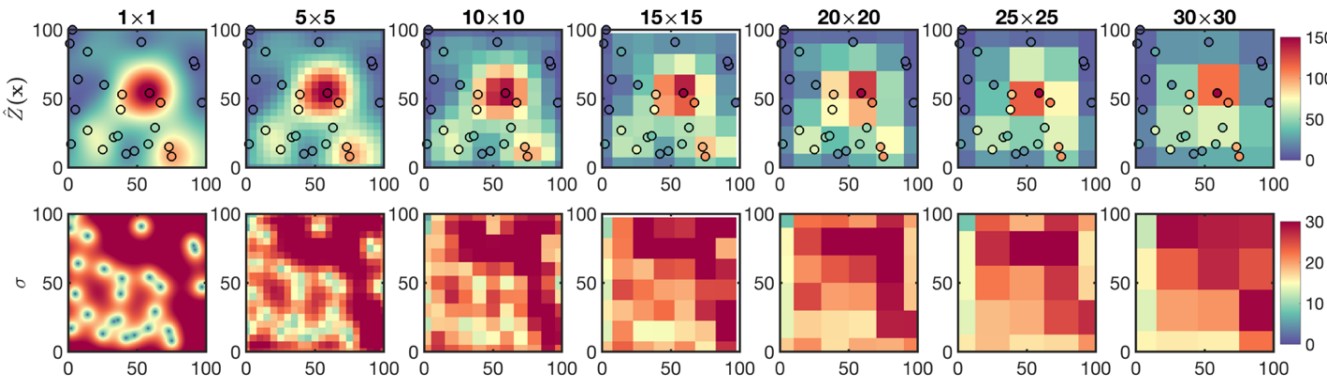

**Figure 6.** (first row) C5Opt outputs convolved with an ideal box kernel with different sizes (1×1 up to 30×30) overlaid by the C5Opt optimum samples. (second row) the associated kriging errors convolved with the same kernel. The coarser the resolution is, the larger the discrepancy between the samples and the estimates is.





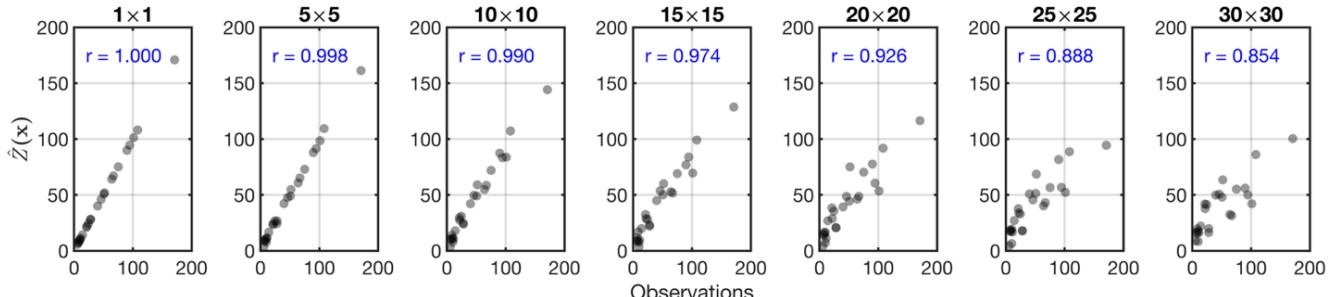

**Figure 7.** Illustrating the problem of spatial scale: comparisons of the kriging estimates at seven different spatial scales with the samples used for the C5opt estimation. The perceived discrepancies are purely due to the spatial representativeness.


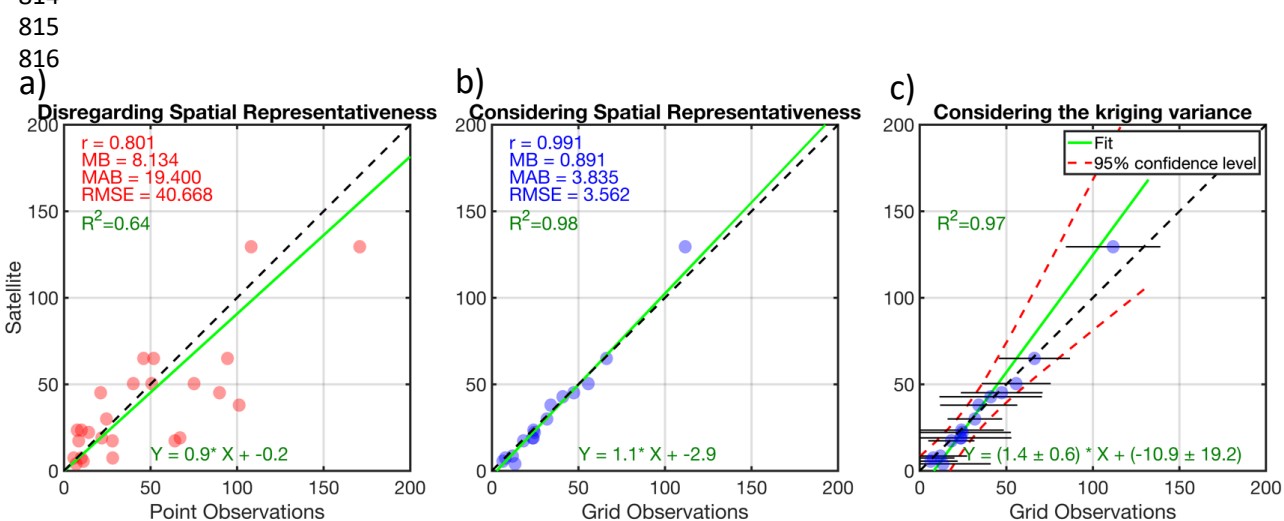

**Figure 8**. (a) the direct comparison of pseudo observations of a satellite observing the C5 case at
30×30 resolution versus the 25 samples used for C5opt. (b) same for y-axis, but the point samples
are transformed to grids using kriging convolved with the satellite spatial response function (ideal
box with 30×30 kernel size). The differences in statistics between these two experiments speak to
the problem of scale. (b) ignores the kriging errors but (c) incorporates them using a Monte Carlo
method. Note that the best linear fit has changed indicating that the consideration of the kriging
variance is critical. MB = mean bias (point minus satellite), MAB = mean absolute bias, RMSE =
root mean square error, $R^2$ = coefficient of determination.





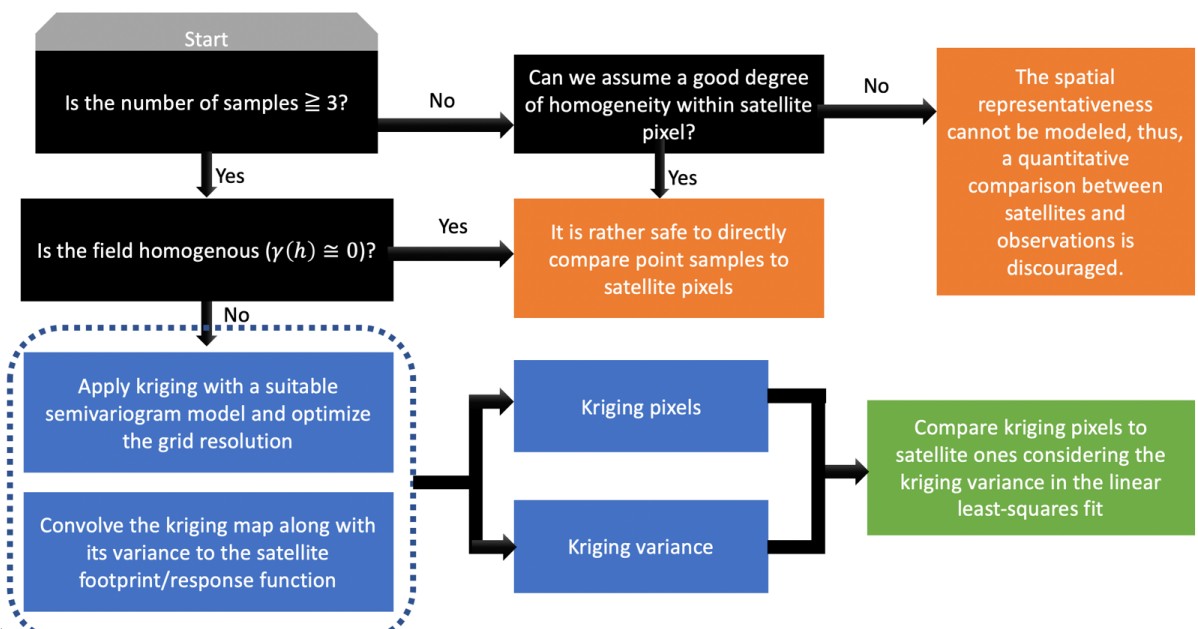

**Figure 9.** The proposed roadmap for transforming pointwise measurements to gridded data in
satellite (model) validation.



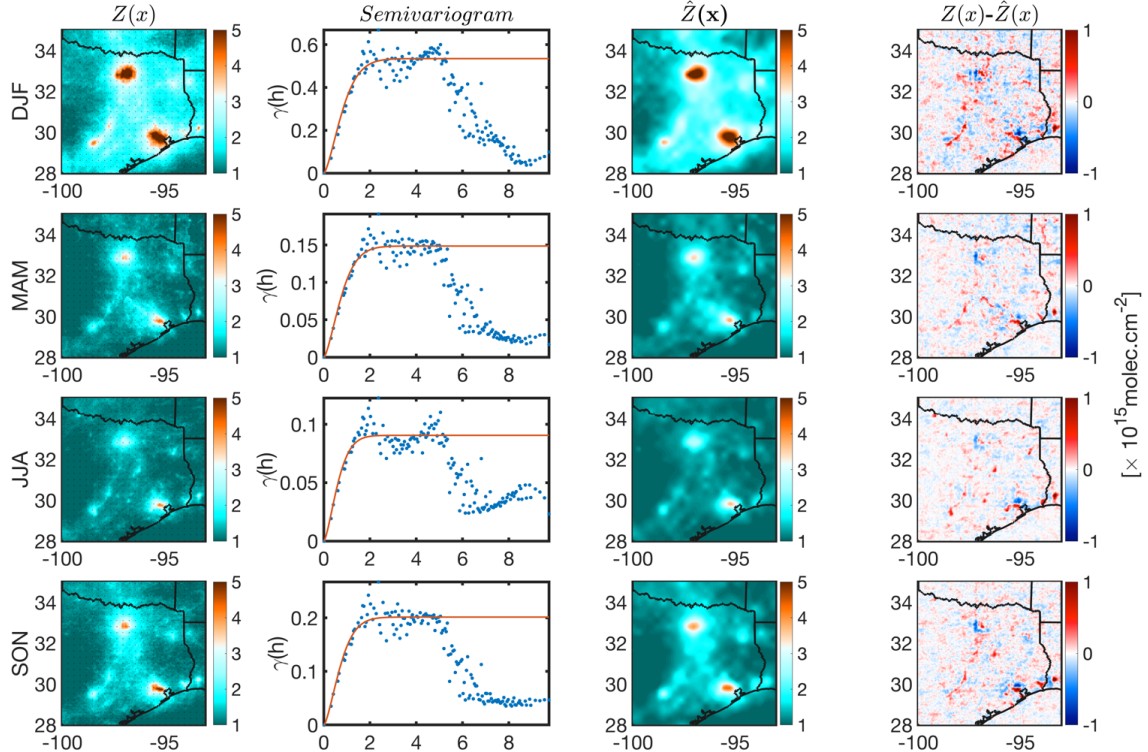

**Figure 10.** (first column) The spatial distribution of TROPOMI tropospheric NO$_2$ columns
oversampled in four different seasons at 3×3 km$^2$ spatial resolution. (second column) The
corresponding semivariogram from samples selected from uniform 30×30 km$^2$ blocks (shown
with black dots in the first column) along the fitted stable Gaussian model (red line). (third
column) the kriging estimates, and (fourth column) their differences with respect to the
observations.




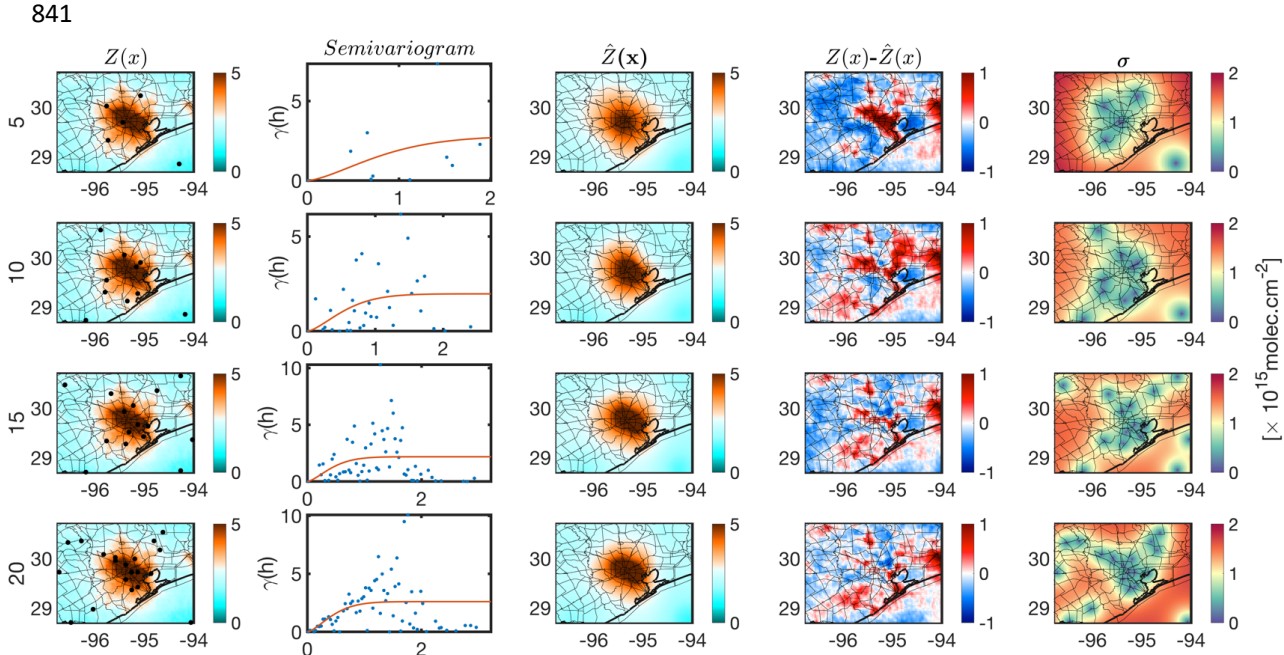

**Figure 11.** Finding an optimum sample tessellation for wintertime over Houston given different
number of spectrometers (5, 10, 15, and 20).


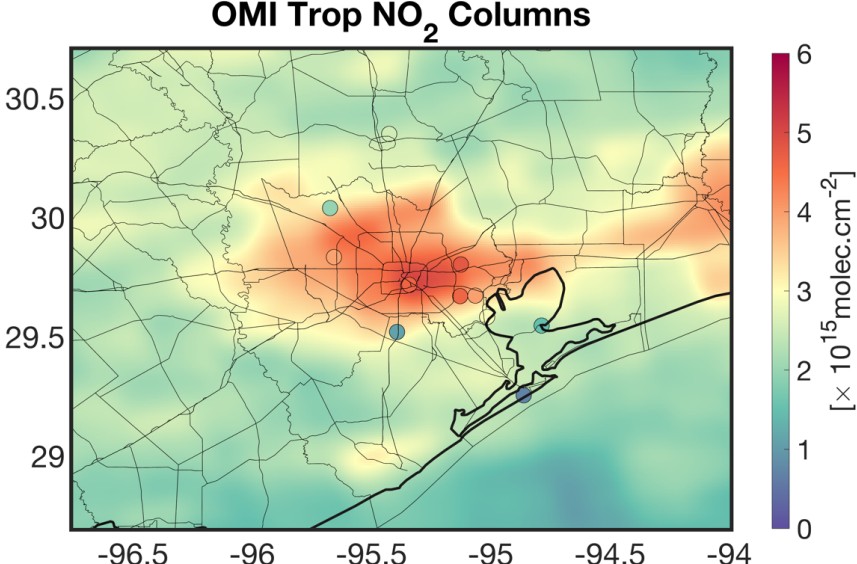

**Figure 12.** The spatial distribution of OMI tropospheric $NO_2$ columns oversampled at the resolution at $20 \times 20$ km$^2$ over Houston in September 2013. The plot is overlaid by surface Pandora spectrometer instrument averaged over the same month. The surface measurements originally measured the total columns, therefore we subtract their values from the stratospheric columns provided by the OMI data ($2.8 \pm 0.16 \times 10^{15}$ molecules cm$^{-2}$) to focus on the tropospheric part.




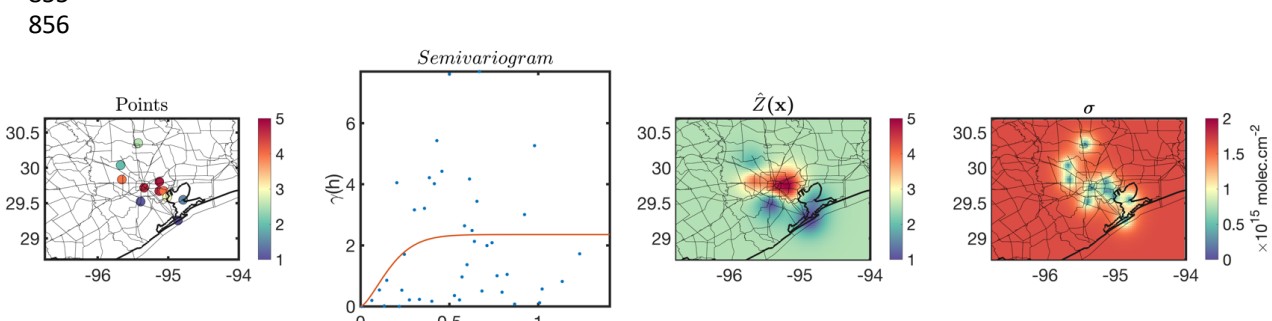

**Figure 13.** The Pandora tropospheric $NO_2$ measurements (made from subtracting the total columns
from the OMI stratospheric $NO_2$ columns) during September 2013, the corresponding
semivariogram, the kriging estimates, and the kriging standard errors. Note that the semivariogram
suggests a large degree of spatial heterogeneity occurring at different spatial scales.





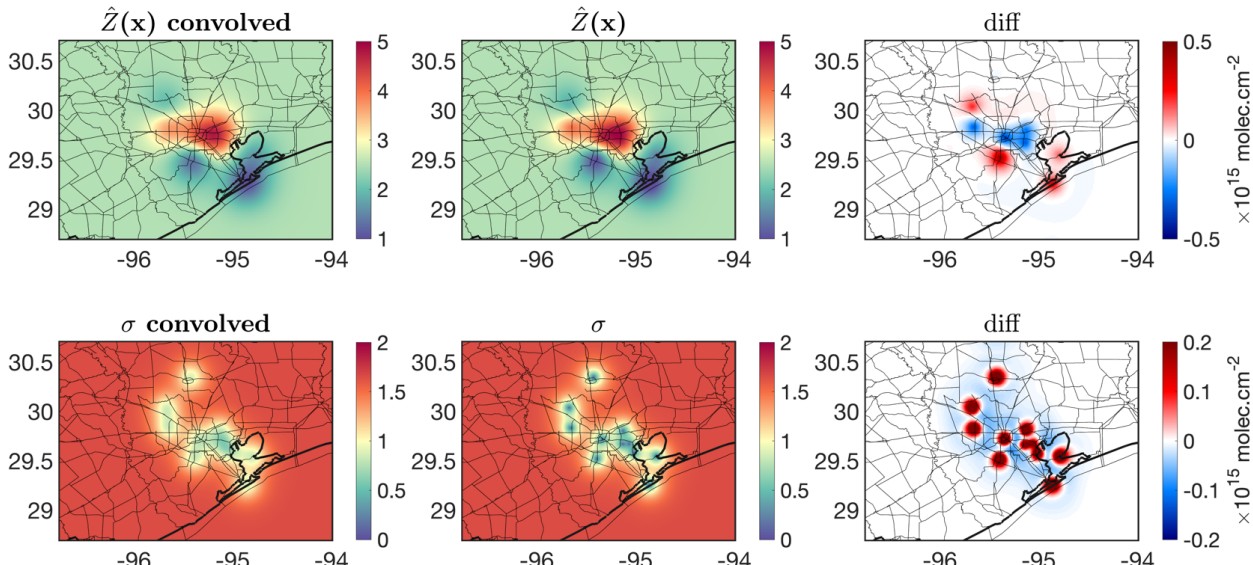

**Figure 14.** Convolving both kriging estimates and errors with the OMI spatial response function formulated in Sun et al. [2018]. The differences against the pre-convolved fields are also depicted.

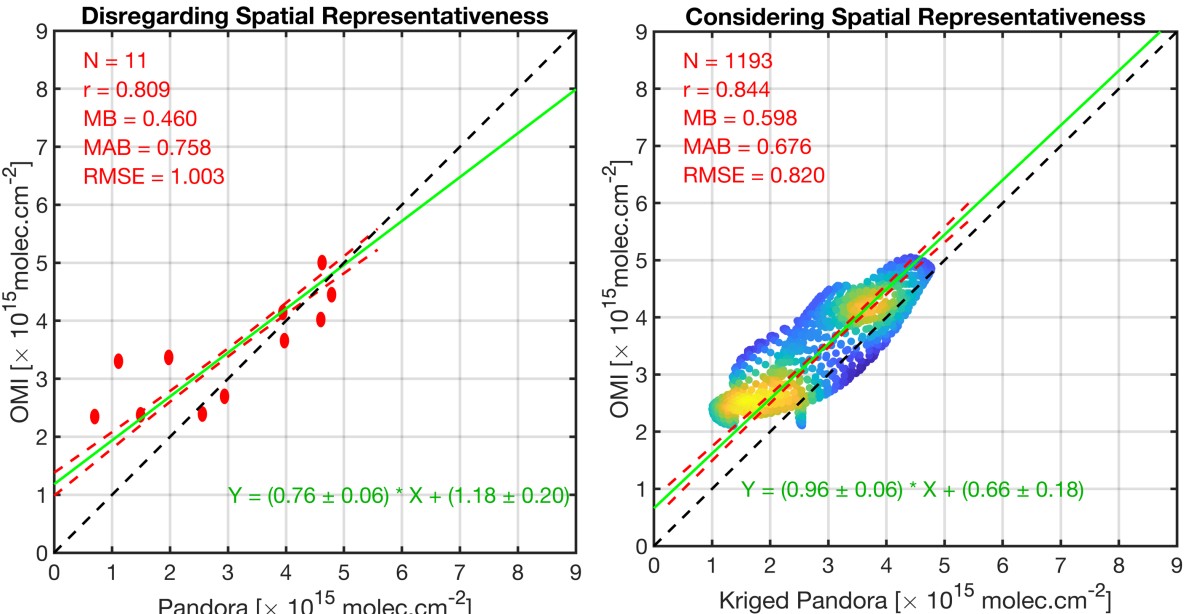

868

**Figure 15.** (left): the direct comparison of OMI tropospheric $NO_2$ columns with 11 pointwise
Pandora measurements in September 2013 over Houston. (right) same for y-axis, but the PSI
measurements are translated to grids using kriging convolved with the OMI spatial response
function. PSI tropospheric $NO_2$ columns are estimated based on subtracting their total columns
from the OMI stratospheric $NO_2$ ones ($2.8\pm0.16 \times10^{15}$ molecules cm$^{-2}$). We only consider kriging
estimates whose errors are below $1.2\times10^{15}$ molecules cm$^{-2}$. The kriging variances are also
considered using the Monte Carlo method on $\chi^2$. The slope has improved after considering the
modeled spatial representativeness. MB = mean bias (OMI vs Pandora), MAB = mean absolute
bias, RMSE = root mean square error.

878