# Peer review of "Dealing with Spatial Heterogeneity in Pointwise to Gridded 2 Data Comparisons"

_Atmospheric Measurement Techniques, 2021_

## Author Comment (AC1)

**General Comments**

The authors introduce the kriging technique, originally from the geostatistics community, to the atmospheric science community. To properly compare point samples (such as Pandora spectrometer) and large satellite pixels (such as Ozone Monitoring Instrument (OMI) pixels), the authors take the following steps: 1) construct a semivariogram that takes account for spatial variances among the point samples, and subsequently produce a kriging estimate based on the point samples (as well as an error estimate) over a 2D grid, and 2) convolve the kriging estimate and error using a spatial response function that represents the large satellite pixel size. The authors first show detailed examples of this process using typical theoretical cases. Finally presented is an actual case comparing NO2 columns from point measurements from Pandora instruments and OMI.

This paper is well organized and fits well in the scope of Atmospheric Measurement Techniques. As this paper introduces a new, useful technique to the atmospheric science community, adding explicit statements to help readers' better understanding will greatly benefit the community. I would recommend the paper for publication in AMT after addressing the specific comments listed below.

**We thank the reviewer for their thoughtful and constructive comments. Our response follows:**

**Specific Comments**

Overall: I suggest using the term "grid box" rather than "grid" when it actually means a grid box.

**Thanks, we changed "grid" to "grid box" whenever it means a cell/pixel throughout the paper.**

Line 75: Studies to downscale satellite pixels using high-resolution model simulations worth to be mentioned, e.g., Kim et al., 2018, and Choi et al., 2010 (already referred to in the text).

Kim, H. C., Lee, S.-M., Chai, T., Ngan, F., Pan, L., and Lee, P.: A conservative downscaling of satellite-detected chemical compositions: NO2 column densities of OMI, GOME-2, and CMAQ, Remote Sensing, 10, 1001, https://doi.org/10.3390/rs10071001, 2018.

Thanks, we included both studies, but it is also important to acknowledge that the downscaling methods heavily rely on the performance of CTMs. In order to characterize the errors associated with the models using measurements, we will need to deal with the spatial heterogeneity issue down the road. We do not think there is a shortcut way to avoid this fundamental problem.

"It is because of this reason that several validation studies resorted to downscaling their relatively coarse satellite observations using high-resolution chemical transport models so that they could compare satellites to spatially finer datasets such as in-situ measurements [Kim et al., 2018; Choi et al., 2020]. Nonetheless, their results largely arise from modeling experiments which might be biased."

Line 115, Eq. (1) and Eq. (2): I think h in g(h) should be boldfaced, as in f(x+h) in the later part of the equation? Although this study deal with isotropic cases only, the length of vector h, i.e., h = |h|, needs to be explicitly defined or mentioned before being used in the text.

**Thanks for the precise comment. Yes, we boldfaced it, and added:**

"Under this condition, the vector of **h** becomes scalar (h = |h|)."

Line 122: Parallel to the above point, I suggest explicitly mentioning that  $\gamma(h)$  here is only dependent on the distance between samples, not direction/angle.

**Addressed above.**

Line 130, Eq. (4) While the authors state that  $a_o$ ,  $b_o$ ,  $c_o=1.5$ , they fit paired samples into the given Gaussian function. Therefore, these coefficients cannot be fixed values. Moreover, red lines in all figures variate as well. Please check.

**Sorry for the confusion, only c0 is a constant. We removed a and b and explicitly mentioned:**

"where *a* and *b* are fitting parameters."

We also removed this part to remove further confusion:

In case of two samples, the semivariogram might be explained by a line with no offsets (i.e.,  $\gamma(h) = a_{\sigma}h$ ) or a constant function ( $\gamma(h) = b_{\sigma}$ ).

Line 140, Eq. (7): Does  $Z_{\circ}$  means  $Z(x_{\circ})$  (true value at  $x_{\circ}$ )? Please define  $x_{\circ}$  and  $Z_{\circ}$ .

**We clarified it in the text:**

",  $x_o$  is the location of estimation,  $x_j$  is the location of samples, , and  $Z(x_j)$  is point data (i.e., samples)."

**,and:**

where  $Z_0$  is point observations ( $Z_o = Z(x_j)$ , j = 1, 2, ..., n),

Line 145, Eq. (9): Do  $\gamma_{j_{1}j_{2}}$ ,  $\gamma_{j_{1}o}$ ,  $\gamma_{oo}$  mean  $\gamma(x_{j_{1}}-x_{j_{2}})$ ,  $\gamma(x_{j_{1}}-x_{o})$ , and  $\gamma(x_{o}-x_{o})$ , respectively? Please define them explicitly in the text.

**Thanks, yes, we clarified it:**

**where $\gamma_{j1j2}$ is the spatial covariance between the point observations and $\gamma_{j10}$ is the spatial covariance of between the observations and the estimation point. The spatial covariance is modeled by a semivariogram.**

Overall Sect.2.1: There are too many subscripts  $_0$  or  $_0$ . The coefficients of the Gaussian function are  $a_0$ ,  $b_0$ , and  $c_0$ . A specific, random point we want to estimate Z is  $x_0$ .  $l_0$  is a constant weight.  $Z_0$  is (probably) the true value at a point  $x_0$ . Some of these subscripts are relevant but others might not. Removing unnecessary 0 subscripts may help readers understand better.

**We are grateful for this precise comment. We got rid of all "o" subscripts and replaced them with "0". We redefined some of the parameters to remove any confusion.**

Line 163-164: The authors take 200 samples and make 100 pairs. However, the maximum number of pairs seems  $_{200}C_2$ , or  $_nC_2$  (n=number of samples), according to the first row of Fig.2 and the second panel of Fig. 13. This is worth mentioned here.

**This comment is a bit unclear to us. From 200 samples, we can create 19900 paired values (i.e., $\binom{200}{2}$ ), the paired values were binned to 100 evenly binned distances depending on the min/max of distances.**

Line 165 and Fig. 1: The authors mention that the semivariograms except C1 fit to the Gaussian function. I am not sure if the semivariogram of C2 is really Gaussian.

**The gaussian function fit to the semivariograms converges to a fixed value (it is a half-bell shape). As for C2, the range tends to be large (~95) making the half-bell converges to a fixed value in much further values beyond the observed distances.**

Line 223, Fig 2, and Fig. 3: The authors mention the relative error of C5 in line 223, but it is not shown in Fig 2 and Fig 3. Locations with large Z values will naturally exhibit larger error values as shown in the fourth columns of Fig. 2 and Fig.3. However, showing relative error in these figures might be meaningful, as reasonably illustrating the plumes is more important than the absolute value of the error in this case.

We agree with the reviewer that targeting a relative error might be more suitable if the goal is to better represent the shape rather than the absolute values. But it is entirely subjective and the primary goal of kriging is to accurately reconstruct the absolute values of Z given discrete samples. Therefore we decided to leave the figures alone.

Line 248: I suggest replacing "realization" with "kriging attempts" for better understanding.

**We replaced it with the suggestion.**

Line 337: The authors mention a two-dimensional super Gaussian spatial response function used in Sun et al. (2018) study. Although this function is a critical component in the actual OMI-Pandora case study (line 473), it is not used with the theoretical cases. Therefore, this sentence is hanging in this line and may confuse readers. It needs a better location in the paper.

Regarding the super Gaussian function: Again, although convolution with this function is a critical component in the actual OMI-Pandora case, no explanation or visual illustration has been made. Instead, only the uniform spatial response function/ideal box kernel is visually illustrated in Fig. 5. I suggest showing a figure showing the convolved C5opt with the super Gaussian function (possibly using various parameters, comparable to Fig. 5) when introducing the super Gaussian function. This figure may go into Supplement.

The reason why we did not use the super Gaussian in the theoretical experiment was to simplify the analysis. We now mentioned this: "For simplicity, we consider  $S = \frac{1}{m^2} J_{m,m}$ ."

Moreover, the super Gaussian function described in Sun et al. [2018] is dependent on the viewing geometry of a real sensor. The utilization of this slit function is more relevant for realworld experiments. The idea behind experimenting with theoretical cases was to show the significance of the problem for simplified scenarios. We now have included an example of the spatial response function for OMI-Pandora comparison:

**Figure S2. An example of the super Gaussian spatial response function described in Sun et al. [2018] for a given pixel over the region of interest.**

Line 339 and Fig.5: The authors suggest  $S[m,n] = 1/(m*n) J_{m,n}$  (J is the matrix of ones) as a uniform spatial response function. Also mentioned is that the panels in Fig. 5 are convolved with an ideal box kernel in the caption of Fig. 5. In summary, the authors mention that "If the spatial response function is assumed to be an ideal box, the resulting grid will represent the average." Putting them all together, "S[m,n] =  $1/(m*n) J_{m,n}$  (J is the matrix of ones)" is the "ideal box kernel", and convolution with an ideal box kernel actually means taking the average within a grid box, which should be explicitly stated here.

**Absolutely right, we mentioned it here:**

**"For simplicity, we consider $S = \frac{1}{m^2} J_{m,m}$ ; this spatial response function results in averaging the values in the grid boxes"**

Line 339-340, Eq. (14): Does  $S_2[m,n]$  in Eq. (14) means the squared [m,n]-th element in the matrix S[m,n] in line 339? If so, please explicitly state it. Also, taking a matrix notation for the matrix S[m,n] will be helpful.

**Yes, we added "where a superscript of 2 denotes squaring."**

**Regarding the notation, the equations (13,14) are a very standard presentation of a convolution process which can be seen in many different image/signal processing books.**

Line 358: Is there a reason for not showing the synthetic satellite measurements (upscaled truth)? I am interested to see how synthetic satellite measurements compare with the kriging estimate in the 30x30 resolution (although it is before converging) and the converged kriging estimate (1x1) resolution convolved into the 30x30 resolution.

**Sure, we now have added the synthetic measurements in the supplementary material (Figure S1).**

Figure S1. The synthetic satellite observations of the field of truth at the resolution of 30×30 pixels.

Regarding the comparison of non-converged kriging at 30x30 and the synthetic observations, we can observe in Figure 5 that the kriging estimate substantially differs from the truth at coarse resolutions indicating that the estimate should compare poorly with the synthetic satellite observations too. The next figure (for reviewer only) shows the kriging estimate at 30x30 grids:

We are reluctant to include this figure in the paper as our analysis in Figure 5 strongly suggests that we should optimize the resolution of kriging map before comparing/convolving with other datasets.

Technical corrections:

Line 130, Eq. (4): In the equation, some subscripts look like alphabet o while others look like number 0. Please make them consistent.

Line 138 & Eq. (10): Also, while the subscript of  $x_0$  in line 138 looks like a number 0, subscript of  $x_0$  in Eq. (10) looks like an alphabet o.

**For consistency, we replaced "o" with "0".**

Fig. 10: Dots showing uniform sampling locations are barely visible. Can they be more visible?

Sure, we reworked the figure with larger dots. Due to the fact that we used a random function, the results are slightly different for the random and stratified random cases compared to the original draft.

New figure:

---

## Author Comment (AC2)

The authors suggested a classical geostatistical approach (using semivariogram and kriging) to deal with spatial heterogeneity in point and grid data comparisons. The value of the approach was also demonstrated using both theoretical and real-world experiments. Overall, this paper is well written and falls into the scope of AMT. The following comments need to be addressed before publishing.

**We thank this reviewer for their positive/generous comments. Our response follows:**

The approach suggested in this manuscript is very useful. However, given the complexity and computational cost, it may not always be practical for other groups to apply. I'm wondering if the authors have any suggestions or comments on this?

**The semivariogram modeling and calculations along with different types of kriging (ordinary, simple, universal, …) have been implemented in many different programming languages. It is extremely fast to get the results such that all of experiments presented in the paper were done on a personal laptop. A couple of useful links on publicly available geostatistical toolboxes:**

**MATLAB:** https://www.omicron.dk/dace.html
 https://www.mathworks.com/matlabcentral/fileexchange/25948-variogramfit
 https://www.mathworks.com/matlabcentral/fileexchange/29025-ordinary-kriging
**Python:** https://geostat-framework.readthedocs.io/projects/pykrige/en/stable/
 https://gmd.copernicus.org/preprints/gmd-2021-301/
**R:** http://www.gstat.org/

The authors suggest that when the sampling is sparse (<3 samples within the field) and a good degree of homogeneity can't be assumed, a quantitative comparison between satellite and observations is discouraged. However, it is common for some ground networks to have sparse data. For example, the TCCON sites are generally far from each other and are often used for satellite evaluation. I'm wondering if the authors have any comments on this.

**Mathematically, it is impossible to gain spatial variance from discrete data if we have fewer than 3 point samples. This is really pure math, and is not a suggestion made by the present study. That means those sparse networks such as TCCON will never be able to provide the information on the spatial distribution of greenhouse gases at the scale of satellite footprints. There are two solutions to this problem: i) we should increase the number of point observations for the satellite validation purpose, and ii) we may use higher spatial resolution airborne data such as MethaneAIR (https://amt.copernicus.org/articles/14/3737/2021/amt-14-3737-2021.html) flying over TCCON stations. These data have sufficiently fine spatial resolution such that you can directly compare them to the point measurements without being too stressed about the problem of scale. The bias-corrected airborne observations then can be upscaled to a satellite footprint and further be compared.  On a brighter note, there are many air quality related campaigns possessing a dense network of spectrometers. Furthermore, EPA provides a large suite of surface observations with relatively high spatial density.**

In Section 2.1, $a_0$ is used multiple times (e.g., equations 4 and 7). Are they the same? To avoid any potential confusion, if they are the same, please explicitly indicate it. Otherwise please use different symbols.

**Thanks for your comment, we have changed it to make sure it won't be confused with a0.**

Figure 1: This change is not necessary, but it would be great if you could make the semivariogram plot for C1 to have the same x-axis labels as the C2-C5.

Line 236 Please briefly explain how did you "classify the domain into four zones using the k-mean algorithm" if possible.

**We simply classify the magnitude of Z(x) using k-means. The inputs are only the magnitudes.**

**We classify the domain into four zones** by running the k-mean algorithm on the magnitudes of $Z(x)$

Line 276: Please change "distance" to "difference" or "bias" to avoid any potential confusion.

**Sure, we changed it to difference.**

---

## Author Comment (AC3)

'Dealing with Spatial Heterogeneity in Pointwise to Gridded Data Comparisons' demonstrates a new methodology for comparing point measurements to those that are more spatially representative of an area (e.g., satellite or models) for the purpose of data product validation. This topic takes an important challenge with these types of data comparisons for the purposes of validation and attempts to lay out a solution going forward. The manuscript is very well written and organized and it fits well within the scope of AMT and should be considered for publishing after some minor changes.

**We thanks reviewer for their thoughtful and constructive comments. Our response follows:**

General comments:

• The paper switches back and forth between the nomenclature of using the method in relation to validation vs. data comparisons. While the methodology and conclusions made related to using point measurements for validating grid-like data are completely valid and important, the language in this paper is dismissive of other potential reasons to compare point-like measurements with something like a model or satellite. Therefore, the suggestion is to clarify that this methodology is for validation and not generalize conclusions for data comparisons.

Thanks for your comment. To be able to validate volumetric data (such as satellites and models) using pointwise measurements, we have to compare them. This comparison is apples to oranges. Since all apples-to-oranges comparisons are wrong, we, as scientists must be alert to what is importantly wrong with respect to such a comparison. Validation comes with data comparisons, and data comparisons are meant for validation of a product or a hypothesis. They both are inextricably linked together. So we cannot say that it is okay to compare two different things as long as we do not want to use the quantitative statistics made from the comparison.

- Examples of this type of language include, but are not necessarily limited to:
  - Line 80-81 with the question of whether a 'comparison ever logical' in the right case the comparison could be logical when trying to learn how satellite data can be interpreted in relation to a ground-based measurements

**To account for this, we added: If one compares a grid box to a point sample (i.e., apples to oranges)**

**And:**

can the average of the spatial distribution of the underlying compound be represented by a single value measured at a subgrid location?

• Line 551: change 'comparisons' to something along the lines of validation

**"The validation against point measurements can be carefully conducted in the following steps:"**

• Line 572: 'point-pixel comparisons' should say pixel validation with point measurements or something along that line.

We changed it to: "validating satellites/models using pointwise measurements"

• Figure 9: in the right orange box change 'comparison between satellites and observations' to 'validation of satellites (models) with point observations'

**We changed it:**

• Could the authors comment on the reality of ground-based networks that could actually contribute to satellite/model validation with this methodology? Does the required observational density exist anywhere? What are some paths forward/recommendations?

Several DISCOVER-AQ campaigns such as Texas 2013 and Colorado 2014 possess an adequate number of Pandora samples (~10-12) to validate sensors like OMI or GOME. We are obviously in better shape when it comes to model validation. EPA provides a very dense network of NO2, O3, and PM2.5 in the U.S. providing a sufficient number of samples to extract a good degree of spatial variance. It is also desirable to compare surface NO2 concentrations to satellite columns which can be effortlessly done with this method. Indeed, there are networks that do not meet the requirement on the availability of the observations at small scales. Off the top of our head, the FTIR HCHO network is one of them. For those cases, we recommend oversample columns from a high-resolution sensor to gauge the level of spatial heterogeneity in the field at the resolution of a coarser satellite/model. That level of heterogeneity should be accounted for in the comparisons made between pixel vs points; for example, over a city, a larger deviation between point and pixel from the identity line can partly be explained by a larger level of spatial heterogeneity suggested by the oversampled field from the fine sensor. Our method won't be applicable for this case, but there are ways to articulate that the problem of scale exists and correlates with the discrepancies we observe between the two different datasets.

• Section 4: Why is v3.0 OMI data using instead of the most up to date v4.0?

Lamsal, L. N., Krotkov, N. A., Vasilkov, A., Marchenko, S., Qin, W., Yang, E.-S., Fasnacht, Z., Joiner, J., Choi, S., Haffner, D., Swartz, W. H., Fisher, B., and Bucsela, E.: Ozone Monitoring Instrument (OMI) Aura nitrogen dioxide standard product version 4.0 with improved surface and cloud treatments, Atmos. Meas. Tech., 14, 455–479, https://doi.org/10.5194/amt-14-455-2021, 2021.

Our first attempt to validate the tropospheric OMI NO2 from Goddard was to use the most updated version. Unfortunately this new dataset showed some unrealistic distributions of NO2 at the northeast of the domain. We closely look at SCD and AMF variables and realized that the artifact was not induced by SCD but rather it originated from the shape factors. Unfortunately at that point, we did not have the CTM outputs used in Souri et al. [2016] to rectify the issue. Luckily, the older version of OMI data whose AMFs had already been recalculated using high resolution WRF-CMAQ model presented in Souri et al., [2016] were available at that time. So we resorted to using that version which was free of the artifact. Regardless of the version, OMI shows incredible accuracy over the region which explains why the study of Souri et al. [2016] was successful at constraining emissions.

**Specific comments:**

• The first sentence in the abstract implies that the two communities have zero realization of the point vs grid problem, which is not true. This is a known problem with the lack of an easy solution. Please consider rephrasing.

**We changed the subject to be less direct and used "assume" to be less harsh: "Most studies on validation of satellite trace gas retrievals or atmospheric chemical transport models assume that pointwise measurements...".**

• Lines 28-29: This study demonstrates a method but it doesn't actually prove that the only available method 'must taking kriging variance...'etc. State what the paper demonstrates without implying there is no other alternative to this exact method.

**We agree. We added: "This study suggests that satellite validation procedures using the present method must take kriging variance and satellite spatial response functions into account."**

• Line 50: consider adding some clarity to the hypothetic scenario by adding after 'atmospheric model' the phrase 'simulating CO2 emissions'...

**Sure, we added: "simulating CO2 concentrations".**

• Line 191: is there a reference for the terminology of 'the sill'?

**Yes, we added "[Chilès and Delfiner, 2009]"**

• Paragraph spanning lines 232-239: Use the word stratified somewhere to connect to the second row of the figure.

**We added: "As a remedy, it might be advantageous to group the domain into similar zones and randomly sample from each, which is commonly known as stratified random selection"**

• Line 385: add that this is also a roadmap for model evaluation as well.

**Added.**

• Line 414: add a reference for the length scale of NO2 if connecting it to the range found in those months.

**Sure, "These numbers strongly coincide with the seasonal lifetime of NO2 [Shah et al., 2020]"**

• Lines 433-440: More clearly explain the rational more clearly between 10 vs. 15 vs. 20. In the figure alone it isn't all that clear. Was the choice of 15 quantified somehow as the best option or was it subjective?

**We accounted the cost of having more spectrometers. To clarify:**

"The difference between kriging estimate and the TROPOMI observations using 20 samples does not substantially differ in comparison to the one using 15 samples. Therefore, to keep the cost low, a preferable strategy is to keep the number of spectrometers as low as possible while achieving a reasonable accuracy. Based on the presented results, the optimized tessellation using 15 samples is preferred among others because it achieves roughly the same accuracy as the one with 20 samples."

• Line 449: Should be Herman et al. 2009

**Fixed.**

• Line 454-456: what resolution is OMI oversampled to?

**0.2°. We had mentioned that:**" Following Sun et al. [2018], we oversample high quality pixels in the month of September 2013 over Houston at 0.2° resolution."**

• Lines 456-457 and Figure 12 caption. The wording states that the total column was subtracted from the stratospheric column, when it is the stratospheric column that should be subtracted from the total column. Please fix this wording.

**Indeed, we fixed it.**